# Ensemble flood simulation for a small dam catchment in Japan using nonhydrostatic model rainfalls. Part 2: Flood forecasting using 1600 member 4D-EnVAR predicted rainfalls.

Kenichiro Kobayashi[1], Le Duc[2,6], Apip[3], Tsutao Oizumi[2,6] and Kazuo Saito[4,5,6]

[1]Research Center for Urban Safety and Security, Kobe University, 1-1 Rokkodai-machi, Nada-ku, Kobe, 657-8501, Japan
[2]Japan Agency for Marine-Earth Science and Technology (JAMSTEC), Yokohama, Japan
[3]Research Centre for Limnology, Indonesian Institute of Sciences (LIPI), Bogor, Indonesia
[4]Japan Meteorological Business Support Center, Tokyo, Japan
[5]Atmosphere and Ocean Research Institute, The University of Tokyo, Kashiwa, Japan
[6]Meteorological Research Institute, Tsukuba, Japan

*Correspondence to*: Kenichiro Kobayashi (kkobayashi@phoenix.kobe-u.ac.jp)

**Abstract.** This paper is a continuation of the authors' previous paper (Part 1) on the feasibility of ensemble flood forecasting for a small dam catchment (Kasahori dam; approx.70 km$^2$) in Niigata Japan using a distributed rainfall-runoff model and rainfall ensemble forecasts. The ensemble forecasts were given by an advanced data assimilation system, a four-dimensional ensemble variational assimilation system using the Japan Meteorological Agency non-hydrostatic model (4D-EnVAR). A noteworthy feature of this system was the use of a very large number of ensemble members (1600), which yielded a significant improvement in the rainfall forecast compared to Part 1. The ensemble flood forecasting using the 1600 rainfalls succeeded in indicating the necessity of emergency flood operation with the occurrence probability and enough lead time (e.g., 12 hours) with regard to this extreme event. A new method for dynamical selection of the best ensemble member based on the Bayesian reasoning with different evaluation periods is proposed. As the result, it is recognized that the selection based on Nash Sutcliffe Efficiency does not provide an exact discharge forecast with several hours lead time, but it can provide some trend in the near future.

## 1 Introduction

Flood simulation driven by ensemble rainfalls has attracted more attention in recent years with a lot of useful information that ensemble flood forecasts can provide in flood control such as forecast uncertainty, probabilities of rare events, and potential flooding scenarios. In the Japanese case, it is considered that the ensemble rainfall simulation with a high resolution (2 km or below) is desirable since extreme rainfall often takes place due to mesoscale convective systems and the river catchments are not as large as continental rivers; even the largest Tone River Basin, is around 17000 km$^2$.

A good review of ensemble flood forecasting using medium term global/European ensemble weather forecasts (2-15 days ahead) by numerical weather prediction (NWP) models can be found in Cloke and Pappenberger (2009). In much of their

review, the resolution of NWP model is relatively coarse (over 10 km), the number of ensembles is moderate (10-50) and the target catchment size is often large (e.g., Danube River Basin). They basically reviewed global/European ensemble prediction systems (EPS) but also introduced some researches on regional EPS nested into global EPS (e.g., Marsigli et al. 2001). They stated that "One of the biggest challenges therefore in improving weather forecasts remain to increase the resolution and identify the adequate physical representations on the respective scale, but this is a source hungry task".

Short-term flood forecasting (1-3 day) based on ensemble NWPs is gaining more attention in Japan. Kobayashi et. al. (2016) dealt with an ensemble flood (rainfall-runoff) simulation of a heavy rainfall event occurred in 2011 over a small dam catchment (Kasahori Dam; approx. 70 km$^2$) in Niigata, central Japan, using a rainfall-runoff model with a resolution of 250 m. Eleven-member ensemble rainfalls by the Japan Meteorological Agency nonhydrostatic model (JMA-NHM; Saito et al. 2006) with horizontal resolutions of 2 km and 10 km were employed. The results showed that, although the 2 km EPS reproduced the observed rainfall much better than the 10 km EPS, the resultant cumulative and hourly maximum rainfalls still underestimated the observed rainfall. Thus, the ensemble flood simulations with the 2 km rainfalls were still not sufficiently valid and a positional lag correction of the rainfall fields was applied. Using this translation method, the magnitude of the ensemble rainfalls and likewise the inflows to the Kasahori Dam became comparable with the observed inflows.

Other applications of the 2 km EPS, which permit deep convection on some level, can be found in for example Xuan et al. (2009). They carried out an ensemble flood forecasting at the Brue catchment, with an area of 135 km$^2$, in southwest England, UK. The resolution of their grid based distributed rainfall-runoff model (GBDM) was 500 m and the resolution of their NWP forecast by the PSU/NCAR mesoscale model (MM5) was 2 km. The NWP forecast was the result of downscaling of the global forecast datasets from the European Centre for Medium-range Weather Forecasts (ECMWF). Fifty members of the ECMWF EPS and one deterministic forecast were downscaled. Since the original NWP rainfall of a grid average still underestimates the intensity compared with rain-gauges, they introduced a best match approach (location correction) and a bias-correction approach (scale-up) on the downscaled rainfall field. The results showed that the ensemble flood forecasting of some rainfall events are in good agreement with observations within the confidence intervals, while those of other rainfall events failed to capture the basic flow patterns.

Likewise in Europe, Hohenegger (2008) carried out the cloud-resolving ensemble weather simulations of the August 2005 Alpine flood. Their cloud resolving EPS of 2.2 km grid space included the explicit treatment of deep convection and was the result of downscaling of COSMO-LEPS (10km resolution driven by ECMWF EPS). Their conclusion was that despite the overall small differences, the 2.2 km cloud resolving ensemble produces results as good as and even better than its 10km EPS, though the paper did not deal with the hydrological forecasting. Another paper which dealt with cloud resolving ensemble simulations can be found in Vie et al. (2011) for Mediterranean heavy precipitation event. Their ensemble weather simulation model resolution was 2.5 km by AROME from Meteo-France which uses ALADIN forecast for lateral boundary condition (10km resolution), thus the deep convection was explicitly resolved. We can recognize from these researches that

the European researchers especially around mountain region have been farsighted from early days for the importance of these cloud resolving ensemble simulations.

While in Japan, Yu et al. (2018) have also used a post-processing method using the spatial shift of NWP rainfall fields for correcting the misplaced rain distribution. Their study areas are Futatsuno (356.1 km$^2$) and Nanairo (182.1 km$^2$) dam catchments of the Shingu River Basin, in Kii Peninsula, Japan. The resolution of the ensemble weather simulations were 10 km and 2 km by JMA-NHM, which is similar to the downscaling EPS in Kobayashi et al. (2016) but for a different heavy rainfall event in west central Japan caused by a typhoon. The results showed that the ensemble forecasts produced better results than the deterministic control run forecast, although the peak discharge was underestimated. Thus, they also carried out a spatial shift of the ensemble rainfall field. The results showed that the flood forecasting with the spatial shift of the ensemble rainfall members was better than the original one, likewise the peak discharges more closely approached the observations.

Recently, as a further improvement upon the 2 km downscale ensemble rainfall simulations used by Kobayashi et al. (2016), Duc and Saito (2017) developed an advanced data assimilation system with the ensemble variational method (EnVAR) and increased the number of ensemble members to 1600. This new data assimilation system was aimed to improve the rainfall forecasts of the 2011 Niigata-Fukushima heavy rainfall event. The torrential rain of this event occurred over the small area along the synoptic scale stationary front (for surface weather map, see Fig. 1 of Kobayashi et al. 2016). Saito et al (2013) found that the location where intense rain concentrates varied to small changes of the model setting, thus the position of the heavy rain was likely controlled by horizontal convergence along the front, rather than the orographic forcing.

Since the new EPS produced better forecasts of the rainfall fields, in this study, as a Part 2 version of Kobayashi et al. (2016), we applied those 1600 ensemble rainfalls to the ensemble inflow simulations to Kasahori Dam. In the series of Part 1 and 2, we intentionally have chosen a rainfall-runoff model whose specification is quite close to those runoff models used in many governmental practices of Japanese flood forecasting to see the usefulness of 1600 ensemble rainfalls. Our objective is to assess impact of the improvement of the rainfall forecast over the large area around Kasahori dam on the streamflow forecast for the Kasahori dam. In Part 1 the technique of positional lag correction has been applied to match the rainfall forecasts with the observations to have a better hydrological forecast. This technique is hard to be applied in real-time flood forecasting since rainfall observations are unknown and there exist a lot of potential positional lag vectors to choose. Statistically the positional lag vector should respond to the local orographic features but it may vary depending on the synoptic condition on the day and model forecast errors in a specific event. Thus the positional lag vector for one extreme rainfall event basically cannot be applied to other extreme event as is. The new EPS is expected to remove the use of such technique.

In addition, the very large number of ensemble members, which is 10 to 20 times larger than the typical number of ensemble members currently run in operational forecast centers, poses new issues needed to solve in computation and interpretation. First, regional forecast centers may not afford running 1600 hydrological forecasts in real-time and a method to choose the most important members may be helpful. Such kind of method is known as ensemble reduction in ensemble

forecast (Molteni et al., 2011; Montani et al., 2011; Hacker et al., 2011; Weidle et al., 2013; Serafin et al., 2019), which is built upon cluster analysis when observations are not used as a guidance for selection. However, our problem is more interesting where we can access the observations at the first few hours and ensemble reduction should make use of these past observations in selecting important members. Second, it is more challenging to interpret the result when temporal and spatial uncertainties are realized more distinct now. Without taking such uncertainties into account, the ensemble forecasts are easily to be considered as useless.

The organization of this paper is as follows. Section 2 describes the new mesoscale EPS, its forecast and rainfall verification results. Section 3 describes the rainfall-runoff model for explaining the changes in the model parameters. Results are shown in Section 4. In Section 5, concluding remarks and future aspects are presented.

## 2 Mesoscale ensemble forecast

### 2.1 Ensemble prediction system

An advanced mesoscale EPS was developed and employed to prepare precipitation data for the rainfall-runoff model. The EPS was built around the operational mesoscale model JMA-NHM for its atmospheric model as the downscale EPS conducted by Saito et al. (2013). In this study, a domain consisting of $819 \times 715$ horizontal grid points and 60 vertical levels was used for all ensemble members. This domain had a grid spacing of 2 km and covered the mainland of Japan. With this high resolution, convective parameterization was switched off. Boundary conditions were obtained from forecasts of the JMA's global model. Boundary perturbations were interpolated from forecast perturbations of the JMA's operational one-week EPS as in Saito (2013).

To provide initial conditions and initial perturbations for the EPS, a four-dimensional, variational-ensemble assimilation system (4D-EnVAR-NHM) was newly developed, in which background error covariances were estimated from short-range ensemble forecasts by JMA-NHM before being plugged into cost functions for minimization to obtain the analyses (Duc and Saito, 2017). If the number of ensemble members is limited, ensemble error covariances contain sampling noises which manifest as spurious correlations between distant grid points. In data assimilation, the so-called localization technique is usually applied to remove such noise, but at the same time it removes significant correlations in error covariances. In this study, we have chosen 1600 members in running the ensemble part of 4D-EnVAR-NHM to retain significant vertical correlations, which have a large impact in heavy rainfall events like the Fukushima-Niigata heavy rainfall. That means only horizontal localization is applied in 4D-EnVAR-NHM. The horizontal localization length scales were derived from the climatologically horizontal correlation length scales of the JMA's operational four-dimensional, variational assimilation system JNoVA by dilation using a factor of 2.0.

Another special aspect of 4D-EnVAR-NHM is that a separate ensemble Kalman filter was not needed to produce the analysis ensemble. Instead, a cost function was derived for each analysis perturbation and minimization was then applied to obtain this perturbation, which is very similar to the case of analyses. This helped to ensure consistency between analyses

and analysis perturbations in 4D-EnVAR-NHM when the same background error covariance, the same localization, and the same observations were used in both cases. To accelerate the running time, all analysis perturbations were calculated simultaneously using the block algorithm to solve the linear equations with multiple right-hand-side vectors resulting from all minimization problems. The assimilation system was started at 0900 JST July 24th, 2011 with a 3-hour assimilation cycle.

5 All routine observations at the JMA's database were assimilated into 4D-EnVAR-NHM. The assimilation domain was the same as the former operational system at JMA. To reduce the computational cost, a dual-resolution approach was adopted in 4D-EnVAR-NHM where analyses had a grid spacing of 5 km, whereas analysis perturbations had a grid spacing of 15 km. The analysis and analysis perturbations were interpolated to the grid of the ensemble prediction system to make the initial conditions for deterministic and ensemble forecasts.

### 2.2 Rainfall verification

Due to limited computational resource, ensemble forecasts with 1600 members were only employed for the target time of 0000 JST July 29th, 2011. However, deterministic forecasts were run for all other initial times to examine impact of number of ensemble members on analyses and the resulting forecasts. Figure 1 shows the verification results for the 3-hour

precipitation forecasts as measured by the Fraction Skill Score (FSS) (Roberts and Lean, 2008). Given a rainfall threshold and an area around a grid point, which is called a neighborhood, the FSS measures the relative difference between observed and forecasted rainfall fractions in this area. This verification score is used to mitigate difficulty in rainfall verification at grid scale with very high-resolution forecasts in which high variability of rain fields usually makes the traditional scores inadequate due to their requirement of exact match between observations and forecasts at grid points. Thus the solution that

the FSS follows is to consider forecast quality at spatial scales coarser than grid scale by comparing forecasts and observations not at grid points but at neighbourhoods whose sizes are identified with spatial scales. The FSS is normalized to range from 0 to 1 with the value of 1 indicating a perfect forecast and the value of 0 a no-skill forecast which can be obtained by a random forecast.

In Figure 1 we aggregate the 3-hour precipitation in the first and second 12-hour forecasts to increase samples in

calculating the FSS. By this way, robust statistics are obtained but at the same time dependence of the FSS on the leading times can still be shown. Note that an additional experiment with 4D-EnVAR-NHM using 50 ensemble members, which is called 4DEnVAR50 to differentiate with the original one 4DEnVAR1600, was run. The main difference between 4DEnVAR50 and 4DEnVAR1600 is that vertical localization was applied in the former case to generate its ensemble members. As mentioned in the previous section, vertical localization can potentially weaken vertical flows in convective

areas by removing physically vertical correlations. It is very clear from Figure 1 that 4DEnVAR1600 outperforms 4DEnVAR50 almost for all precipitation thresholds, especially for intense rain. Also for high rain-rate, compared to JNoVA, 4DEnVAR1600 forecasts are worse than JNoVA forecasts for the first 12-hour forecasts, which can be attributed to the fact that 4D-EnVAR-NHM did not assimilate satellite radiances and surface precipitation like JNoVA. However, it is interesting to see that 4D-EnVAR-NHM produces forecasts better than JNoVA for very intense rains for the next 12-hour forecasts.

To check reliability of the ensemble forecasts, reliability diagrams are calculated and plotted in Figure 2 for 4DEnVAR1600 and 4DEnVAR50. Since JNoVA only provided deterministic forecasts, reliability diagram is irrelevant for JNoVA. Note that we only performed ensemble forecasts initialized at the target time of 0000 JST July 29th, 2001 due to lack of computational resource to run 1600-member ensemble forecasts at different initial times. Therefore, the same strategy of

aggregating 3-hour precipitation over the first and second 12-hour forecasts in calculating the FSS in Figure 1 is applied to obtain significant statistics. Clearly, Figure 2 shows that 4DEnVAR1600 is distinctively more reliable than 4DEnVAR50 in predicting intense rain. While 4DEnVAR50 cannot capture intense rain, 4DEnVAR1600 tends to overestimate areas of intense rain. The tendency of overestimation of 4DEnVAR1600 becomes clearer if we consider the forecast ranges between 12 and 24 hours. However, for the first 12 hours, 4DEnVAR1600 slightly underestimates areas of light rains. This also

explains why the FSSs of 4DEnVAR1600 are smaller than those of 4DEnVAR50 for small rainfall thresholds in Figure 1.

As examples of the forecasts, Figure 3 shows the accumulated precipitation at the peak period (1200-1500 JST July 29th, 2011) as observed and forecasted by the 4D-EnVAR prediction system (Radar-AMEDAS rainfall data : RA, operational precipitation analysis of JMA based on radar and rain-gauge observations). For comparison, the deterministic forecast initialized by the analysis from JNoVA using the same domain has also been given. Note that the forecast range

corresponding to this peak period is from 12 to 15 hours. Clearly, the deterministic forecast initialized by 4D-EnVAR-NHM outperformed that by the JNoVA, especially in terms of the location of the heavy rain, although the forecast by 4D-EnVAR-NHM tended to slightly overestimate the rainfall amount as verified with the reliability diagrams in Figure 2. This over-estimation can also be observed in the coastal area near the Sea of Japan. Note that a significant improvement was also attained against the former downscale EPS  used in Part 1 (see Fig. 9 of Kobayashi et al. 2016) .

Since it is not possible to examine all 1600 forecasts, the ensemble mean forecast is only plotted in the bottom right of Figure 3. Again, the location of the heavy rain corresponds well with the observed location, as in the case of the deterministic forecast, but the ensemble mean precipitation is smeared out as a side effect of the averaging procedure. Therefore, the ensemble mean should not be used into our hydrological model as a representative of the ensemble forecast. Rather than that, all ensemble precipitation forecasts should be fed into the hydrological model to avoid rainfall distortion

caused by averaging in addition to a faithful description of rainfall uncertainty. Of course with 1600 members this causes a huge increase in computational cost and we will try to reduce this burden by testing a suitable dynamical selection described later in Section 4. To have a glance to the performance of the ensemble forecast we plot one-hour accumulated precipitation over the Kasahori Dam catchment in time series in Figure 4 (Observed data = Radar-AMEDAS). It can be seen that while the deterministic forecast could somehow reproduce the three-peak curve of the observed rainfall, ensemble members tended

to capture the first peak only. Note that some members showed this three-peak curve, such as the best member, but their number was much less than the number of ensemble members.

## 3 Distributed Rainfall-Runoff Model

The distributed rainfall–runoff (hereinafter DRR) model used in Part 1 was applied again in this paper. See Kobayashi et al. (2016) for the details. The DRR model applied was originally developed by Kojima et al. (2007) and called CDRMV3. As described in the previous section, we intentionally have chosen a rainfall-runoff model whose specification is close to those runoff models used by national/local governments since the purpose is more to investigate the usefulness of 1600 ensemble rainfalls.

The parameters of the DRR model were recalibrated in this study using hourly Radar AMEDAS since the amount of total rainfall for the period (762.8mm) is closer to ground rain-gauge (765.0 mm) (Kobayashi et. al., 2016). The hourly Radar-Composite (RC, radar data) of JMA was also used for another recalibration as a reference since Radar precipitation data is in general the primary source for real time flood forecasting. The total rainfall amount with RC was 568.5 mm which is smaller than the ground rain-gauge (765.0 mm). The recalibrated equivalent roughness coefficients of the forest, the Manning coefficients of the river, and the identified soil-related parameters are described in Table 1 with the parameters in Part 1. The simulated discharge hydrographs by RA and RC; and observation are shown in Figure 5 with RA hyetograph. The duration of the calibration simulation is from 0100 July 28th to 0000 July 31th, 2011 JST.

The Nash Sutcliffe Efficiency (hereinafter NSE: Nash and Sutcliffe, 1970), which is used for the assessment of model performance, is calculated as follows:

$$\text{NSE} = 1 - \frac{\sum_{i=1}^{N}\left\{Q_0^i - Q_s^i\right\}^2}{\sum_{i=1}^{N}\left\{Q_0^i - Q_m\right\}^2} \tag{1}$$

$$Q_m = \frac{1}{N}\sum_{i=1}^{N} Q_0^i \tag{2}$$

where $N$ is the total number of time steps (1 h interval), $Q_0^i$ is observed dam inflow (discharge) at time i, $Q_s^i$ is simulated dam inflow (discharge) at time $i$, $Q_m$ is the average of the observed dam inflows.

In the calibration simulations in Figure 5, the NSEs with RA and RC are 0.686 and 0.743 respectively. Although the NSE with RA is worse than RC, the total rainfall amount with RA is considered more accurate and the 2nd and 3rd discharge peaks seem to be captured better with RA, thus the following discussion will be made basically with the parameters calibrated with RA. Some results with RC will be added as references. The main difference of the parameters between RA and RC is that the surface soil thickness D to hold the rainfall at the initial stage is thicker in RA, which yields the lower discharge in the river.

## 4 Results

In this section, the results of the ensemble flood simulations are shown focusing on two aspects:

(1) We examined whether the ensemble inflow simulations can show the necessity of starting the flood control operations and emergency operations with sufficient lead time (e.g. 12 h).

(2)  We also examined if we could obtain high accuracy ensemble inflow predictions several hours (1-3 h) before the occurrence, which could contribute to the decision for optimal dam operation.

Item (1) provides us with the scenario that we can prepare for any dam operations with enough lead time. Likewise, it may enable us to initiate early evacuation of the inhabitant living downstream of the dam. Item (2) is the target that has been attempted by researchers of flood forecasting. If we could forecast the inflow almost correctly several hours before the occurrence, it could help the dam administrator with the decision for actual optimal dam operations.

### 4.1 Probabilistic forecast

Item (1) is considered first herein. Figure 6 shows the comparisons of the hydrographs of (a) 11 discharge simulations in Part 1, (b) same 11 member but with a positional shift in Part 1, (c) 50 discharge simulations with 4D-EnVAR-NHM and calibrated parameters by RA, (d) 1600 discharge simulations with 4D-EnVAR-NHM and parameters by RA, and (e) 1600 discharge simulations with 4D-EnVAR-NHM and parameters by RC. Note that the duration of the 4D-EnVAR-NHM ensemble weather simulation is 30 hours from 0000 July 29th to 0700 July 30th JST, but the ensemble flood simulation is carried out only for 24 hours from 0300 July 29th to 0300 July 30th, 2011 JST since we consider that JMA-NHM uses the first 3 hours to adjust its dynamics. The result in Figure 6 (d) and (e) shows that, except for the third peak, the 1600 ensemble inflows can encompass the observed ~~rainfall~~ runoff within the 95 % confidence bound, which was not realized in Part 1 with 11 downscale ensemble rainfalls of 2 km resolution (Figure 6 (a)). In other words, the extreme rainfall intensity of the event can be reproduced by the ensemble members with 1600 4D-EnVAR-NHM on some level. By comparing (d) with (e), it is recognized that the 95 % confidence and interquartile bound of (d) is narrower than (e), thus the prediction with the parameters calibrated with RA, a physically more accurate rainfall, can reduce the uncertainty of the prediction probably because of the better physical meaning in the parameters. It is considered also that the ensemble mean and median values capture the overall trend of the observations on some level.

Likewise, comparing Figures 6 (c) and (d), we can recognize that the simulated discharges by 50 ensemble rainfalls of 4D-EnVAR-NHM does not encompass the observation within the range unlike the 1600 ensembles, thus 50 ensemble discharges cannot be used for the forecasting as they are.

Figure 7 shows the probability that the inflow discharge is beyond 140 $m^3$ $s^{-1}$ (hereinafter expressed as "q > 140", where q is the discharge), the threshold value for starting the flood control operations. The figure considers the temporal shift of the ensemble rainfalls, i.e., temporal uncertainty due to the imperfect rainfall simulation. In the figure, 0-hour uncertainty means that we only considered discharges at time t to calculate probability, while 1-hour uncertainty means that we considered the discharges at t-1, t, t+1 to calculate probability and 2-hour means that we considered the discharges at t-2, t-1, t, t+1, t+2 to calculate probability. The 3- and 4-hour uncertainties were calculated in the same way. It becomes clear from the figure that the starting time of q > 140 is likely at around t = 1000 July 29th JST, where all curves cross, while the ending time is likely at t = 1900 JST, where all curves cross again. Before and after the cross points there are jumps in the probabilities. In other words, the forecast can indicate that the situation of q > 140 would take place after 10 hours from the beginning of

forecasting with the probability of around 50 %. We consider that this is a very valuable information for the users of the ensemble forecast.

On the other hand, the emergency operation was undertaken in the actual flood event. In the emergency operation, the dam outflow has to equal the inflow to avoid dam failure as the water level approaches overtopping of the dam body. As written in Part 1, when the reservoir water level reaches EL 206.6 m, an emergency operation is undertaken, and the outflow is set to equal the inflow. As the Height-Volume (H-V) relationship of the dam reservoir was not known during the study, we judged the necessity of the emergency operation by whether the cumulative dam inflow was beyond the flood control capacity of 8700000 $m^3$. Actually, the flood control capacity had not been previously filled during regular operations more than the estimation given herein, since the dam can release the dam water by natural regulation. However, again, since we do not know some of the relationships to calculate the dam water level, the judgement is done based on whether the cumulative dam inflow exceeds the flood control capacity.

Figure 8 shows the cumulative dam inflows of all the ensemble simulations starting from 0300 July 29th, 2011 JST, as well as 95% confidence and interquartile bound, the mean, median and observed cumulative inflow with the flood control capacity ( (a) with parameters by RA, (b) with parameters by RC). Figure 8 (a) shows that the mean of the cumulative dam inflows underestimates the observation, while in Figure 8 (b) the mean was roughly similar to the observation. Thus, it is considered that many of the 1600 ensemble rainfalls still underestimates the total rainfall amount compared with RA rainfall, though the observed cumulative dam inflow is covered within the 95% confidence interval of the ensemble inflows with RA parameters. As a reference, Figure 8 (b) shows that if the hydrological model parameters are calibrated with less rainfall (this case RC), underestimated ensemble rainfalls yield higher discharges which resulted in almost equivalent mean of the ensembles compared with the observation, though this is considered in physical sense less meaning (see also Figure 6 (d) and (e) for the comparison). Figure 9 shows the probability that the cumulative dam inflow exceeds the flood control capacity of 8700000 $m^3$. The figure indicates that, for instance, the cumulative inflow would exceed flood control capacity after 12 hours from the start of the forecast with the probability of around 15 % (RA parameters) and 45 % (RC parameters), respectively. In the actual event, the cumulative inflow based on observations and assuming no dam water release, would exceed the flood control capacity between 1200 and 1300 July 29th, 2011 JST. Around that interval, the exceedance probability of the forecast is 15-30 % (RA parameters) and 35–55 % (RC parameters). Until around this time, the forecast shows a ~~slight~~ delay in the estimate of the cumulative dam inflow. In the end, the forecast shows that the flood control capacity will be used up with the probability of more than 90 % with regard to this flood event. Thus, we consider this information is very useful as it can inform the inhabitant downstream of the dam to evacuate.

### 4.2 Selection of the best members

Figure 10 shows all ensemble members, the 50 best ensemble members out of 1600 ensembles selected based on NSE > 0.33, and observations. The figure shows that the selected 50 members reproduce the observations well. In some of the selected members, even the 3rd peak is reproduced. In the case where the 3rd peak is reproduced, the inflow hydrographs are

beyond the 95 % confidence interval. Figure 11 shows the catchment average rainfalls of the 50 best ensemble inflow simulations. The black line is the observed gauge rainfall, the blue line is the Radar-AMeDAS (operational precipitation analysis of JMA based on radar and rain gauge observations), the green line is the Radar-Composite, while the grey lines are the 50 rainfalls for the best ensemble discharges. The rainfalls from the best 50 ensemble inflow simulations resemble those

of the Radar-AMeDAS.

Clearly, the flood forecasting becomes very useful if we could just select the best ensemble members in advance. Logically, this is impossible since we only know the best members after knowing the observations which enable us to compute verification scores like NSE. This raises the question whether or not the best ensemble members can be inferred from the partial information provided by the observations at the first few hours. It is easy to see that the answer should be

negative due to nonlinearity of the model and the presence of model error: the best marching at the first few hours is almost certainly not the best marching over all forecast ranges. However, it is obvious that the observations at the first few hours have a certain value which can help to reduce uncertainty in the ensemble forecast if we could incorporate this information into the forecast.

This procedure has already been well-known under the name "data assimilation" in which we assimilate the observations

at the first few hours to turn the prior probabilistic density function (pdf) given by the short-range forecasts into the posterior pdf given by the analysis ensemble (Kalnay, 2003; Reich and Cotter, 2015; Fletcher, 2017). Thus, if we know the observations at the first few hours, we should assimilate these data to replace the short-range ensemble forecasts by the ensemble analyses at these hours, then run the model initialized by the new ensemble to issue a new ensemble forecast. As a result, we should replace the definition of the best members based on verification scores to a more appropriate one based on

the posterior pdf. Here, we identify the best members with the most likely members. Clearly, if we assume the posterior pdf is unimodal, the best members should be the members clustering around the mode of this pdf, which is also the analysis. However, it is not clear how to identify the best members if this pdf is multimodal.

To overcome this problem, we will use the mathematical framework settled up by particle filter (Doucet et al., 2001, Tachikawa et al., 2011). Let us denote the short-range forecasts by $\mathbf{x}_1$ to $\mathbf{x}_K$ where $K$ is the number of ensemble members.

The short-range ensemble forecast therefore yields an empirical pdf given by the sample $(\mathbf{x}_i, w_i^{pre} = 1/K)$ with $w_i^{pre}$ denoting the equal weight for the $i$-th member

$$p_X(\mathbf{x}) = \sum_{i=1}^{K} w_i^{pre} \delta(\mathbf{x} - \mathbf{x}_i) = \sum_{i=1}^{K} \frac{1}{K} \delta(\mathbf{x} - \mathbf{x}_i). \qquad (3)$$

Using this prior pdf as the proposal density, the posterior pdf has the following form

$$p_X(\mathbf{x}|\mathbf{y}) = \sum_{i=1}^{K} w_i^{post} \delta(\mathbf{x} - \mathbf{x}_i) = \sum_{i=1}^{K} \frac{p_Y(\mathbf{y}|\mathbf{x}_i)}{\sum_{j=1}^{K} p_Y(\mathbf{y}|\mathbf{x}_j)} \delta(\mathbf{x} - \mathbf{x}_i). \qquad (4)$$

Here, $p_Y(\mathbf{y}|\mathbf{x}_i)$ denotes the likelihood of the observations $\mathbf{y}$ conditioned on the forecast $\mathbf{x}_i$, and the weight $w_i^{post}$ are the relative likelihoods. Moreover, it can be shown that $p_Y(\mathbf{y}|\mathbf{x}_i)$ is the observation evidence for the ith member (Duc and Saito, 2018). Then applying the model M as the transition model, the predictive pdf is given by

$$p_X(\mathbf{x}|\mathbf{y}, M) = \sum_{i=1}^{K} w_i^{post} \delta(\mathbf{x} - M(\mathbf{x}_i)). \quad (5)$$

This equation shows that the contribution of each member to the predictive pdf is unequal, which differs from the prior pdf (3). While the members with large values of $w_i^{post}$ dominate the predictive pdf, those with very small values of $w_i^{post}$ can be ignored. This suggests that the best members can be identified with the largest values of $w_i^{post}$. Thus, if we sort $w_i^{post}$ in the descending order, the first $N$ weights are corresponding to the first $N$ best ensemble members. In this case, the predictive pdf (5) is approximated by

$$p_X(\mathbf{x}|\mathbf{y}, M) = \sum_{i=1}^{N} \frac{p_Y(\mathbf{y}|\mathbf{x}_i)}{\sum_{j=1}^{N} p_Y(\mathbf{y}|\mathbf{x}_j)} \delta(\mathbf{x} - M(\mathbf{x}_i)). \quad (6)$$

Note that by introducing the notion of the best ensemble members, a substantial change occurs, that is we now work with a unequal weighted sample ($\mathbf{x}_i$, $w_i^{post}$). This should be taken into account in computing statistics like ensemble mean from the best ensemble members.

If the likelihoods have the Gaussian form

$$p_Y(\mathbf{y}|\mathbf{x}_i) \propto \exp\left[-\frac{1}{2}(\mathbf{y} - h(\mathbf{x}_i))^T \mathbf{R}^{-1}(\mathbf{y} - h(\mathbf{x}_i))\right], \quad (7)$$

where h is the observation operator, and $\mathbf{R}$ is the observation error covariance, it is easy to see that the largest weights are corresponding to the smallest weighted root mean square errors (WRMSE)

$$WRMSE_i = (\mathbf{y} - h(\mathbf{x}_i))^T \mathbf{R}^{-1}(\mathbf{y} - h(\mathbf{x}_i)). \quad (8)$$

Therefore, if $\mathbf{R}$ is a multiple of the identity matrix $\mathbf{I}$, the WRMSEs become the RMSEs, which in turn are equivalent to the NSEs. This shows that selection of the best members based on verification scores over the first few hours is in fact selection of the best members based on the relative likelihoods in the posterior pdf. It can also be understood as model selection based on observation evidence (Mackay, 2003).

To check the work of this method of dynamical selection, we attempted to select some of the best members out of the 1600 members several hours in advance of the event based only on NSEs for the discharges. Figure 12(a) shows a result where we selected the best 50 ensemble members (NSE > 0.24) for the first 9 hours from the start of the forecast. In this case, we had a 3-hour lead time towards the observed peak discharge, and the selected 50 members cover the observed discharge after the first 9 hours on some level. The result shows that the ensemble inflow simulations selected can indicate the possibility of rapid increases in the discharge after the 9 hours with a three-hour lead time.

Likewise Figure 12 (b) shows the selected best 50 members (NSE > -0.04) for the first 10 hours (two hours ahead of the observed peak discharge). It is apparent that the result is worse than the previous first 9-hour selection. The ensemble inflow simulations after the 10 hours do not cover the observation well in this case. Figure 12(c) shows the selected best 50 members (NSE > 0.92) for the first 11 hours (1 hour ahead of the observed peak discharge). In this case, the ensemble inflows after the 11 hours could cover the observed peak discharge 1 hour later on some level, although it only has a one-hour lead time.

It is very clear from Figure 12 that the set of the best members varies considerably with the time intervals of available observations. This is because the NSE index is sensitive to the large difference between forecasts and observations. That means, unless we simulate all the discharges of the 1600 members in advance, we may need to run many new members to update this set every time when new observations are available and this causes management of the best members more complicated. To see why this occurs, suppose that we have a member with the sums $\sum_{i=1}^{N-1}\{Q_0^i - Q_s^i\}^2$ are almost zero for the first 1, 2, …, N-1 hours when we only have no rain or light rain during this time. When we consider the next hour to reselect the best members, if the term $\{Q_0^N - Q_s^N\}^2$ becomes very large, this member will suddenly be out of favour despite the fact that it is always selected as one of the best members in all previous selection rounds. However, this large difference may come from spatial and temporal displacement errors of rainfall forecasts and not necessary reflect an inaccurate forecast. This shows that the use of NSE in selecting the best members is quite sensitive to spatial and temporal displacement errors of rainfall. Part 1 of this study is an illustration for impact of spatial displacement errors on forecast performance while Figures 7 and 9 here show the case of temporal displacement errors. On the other hand, NSE of rainfall cannot be used to select the best discharge members since rainfall NSEs of similar values can produce different discharge hydrographs. For example, the catchment average rainfall with NSE of around 0 produces discharges with NSE close to 0.5 and -0.5. The spatial distribution of the rainfall field causes these differences even though the amount of the catchment average rainfalls is the same. Even if the catchment area is small, different patterns in the rainfall field bring different discharge simulations with different NSEs. Furthermore, the error model for rainfall does not follow the Gaussian distribution and a more appropriate distribution like gamma or lognormal should be used. However, such distributions make NSE irrelevant and new verification scores derived from these distributions are needed, which can take the form like FSS. Thus, it is expected that if we can introduce spatial and temporal uncertainty in modelling the likelihood $p_Y(\mathbf{y}|\mathbf{x}_i)$, the predictive pdf (6) could yield a more useful ensemble forecast. However, this requires a lengthy mathematical treatment that is worth to explore in details in a separate study.

## 5 Concluding Remarks and Future Aspects

The study used 1600 ensemble rainfalls produced by 4D-EnVAR which contain various rainfall fields with different rainfall intensities. No post processing such as the location correction of the rainfall field and/or rescaling of rainfall intensity was employed. The ensemble flood forecast using the 1600 ensemble rainfalls in this study has shown that the extremely high amount of observed inflow discharge can be reproduced within the confidence interval, which was not possible by the 11 member downscale ensemble rainfalls used in Part 1. NSE of the best member out of 1600 was 0.72. Likewise, we can calculate the probability of occurrence (e.g. the necessity of emergency dam operations) with the 1600 ensemble rainfalls. Thus, the result of the study shows that the ensemble flood forecasting can inform us that, after 12 hours for example, emergency dam operations would be required with the probability of around 15-30 %, and that the probability would be more than 90 % for the entire flood event, etc. We consider that this kind of information is very useful. For instance, a

warning of dam water release can be issued to the inhabitant in the downstream with enough lead time, if the result obtained in this study is further applicable to other locations and events.

On the other hand, the discharge simulations with similar NSEs until X hours produce different future forecasts after the X-th hour. In other word, we cannot select the best discharge simulation from the NSE only until X hours. Herein lies the problem that, NSEs are quite sensitive to spatial and temporal displacement errors in rainfall. In principle, it is possible to introduce those errors into NSEs in a way similar to FSSs. However, it should be cautious in introducing such errors into NSEs before investigated well, although such type of approach has been used recently in meteorology community. How to incorporate them qualitatively is also a problem to be addressed. Thus, in this sense, the dynamical selection of the best rainfall field from rainfall simulations considering both spatial and temporal displacement errors is required, although this was not addressed here and remains for future work.

*Acknowledgments.* A part of this work was supported by the Ministry of Education, Culture, Sports, Science and Technology as the Field 3, the Strategic Programs for Innovative Research (SPIRE) and the FLAGSHIP 2020 project (Advancement of meteorological and global environmental predictions utilizing observational "Big Data"). Computational results were obtained using the K computer at the RIKEN Advanced Institute for Computational Science (project ID: hp140220, hp150214, hp160229, hp170246, and hp180194). JMA-NHM is available under collaborative framework between MRI and related institute or university. Likewise, the DRR model is available under collaborative framework between Kobe, Kyoto Universities and related institute or university. The JMA's operational analyses and forecasts, radar rain gauge analyses, and radar composite analyses can be purchased at http://www.jmbsc.or.jp/. The rain gauge data were provided by MLIT, Niigata Prefecture and JMA.

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

30

**List of Table**

Table 1. The equivalent roughness coefficient of the forest, the Manning coefficient of the river, and identified soil-related parameters of Part 2 (this paper) and Part 1 (Kobayashi et al., 2016).

| | Forest [m(-1/3)/s] | River [m(-1/3)/s] | D [m] | Ks [ms-1] |
|---|---|---|---|---|
| This paper (RA) | 0.424 | 0.010 | 0.522 | 0.0010 |
| This paper (RC) | 0.170 | 0.005 | 0.234 | 0.0008 |
| Part 1 | 0.150 | 0.004 | 0.320 | 0.0005 |

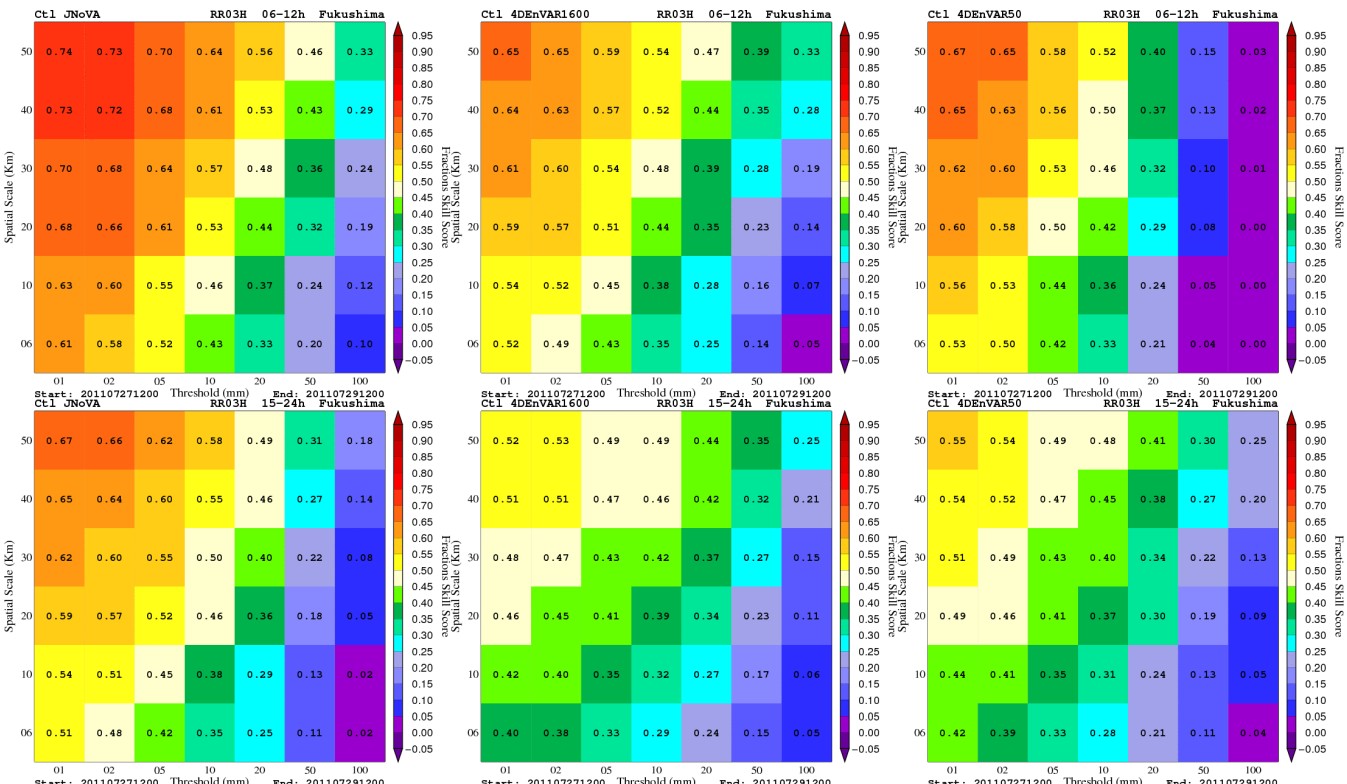

5  **Figure 1. Fraction skill scores of 3-hour precipitation at Fukushima-Niigata from deterministic forecasts initialized by analyses from JNoVA (left), 4D-EnVAR-NHM using 1600 (center) and 50 members (right). These scores are averaged over the period from 2100 JST July 27th to 2100 JST July 29th, 2011. To obtain robust statistics, precipitation is aggregated over the first 12-hour forecasts (valid between 03-12-hour forecast) and the next 12-hour forecasts (valid between 12-24-hour forecasts) as shown in the top and bottom rows, respectively. Note that the first 3-hour precipitation is discarded due to the spin-up problem.**

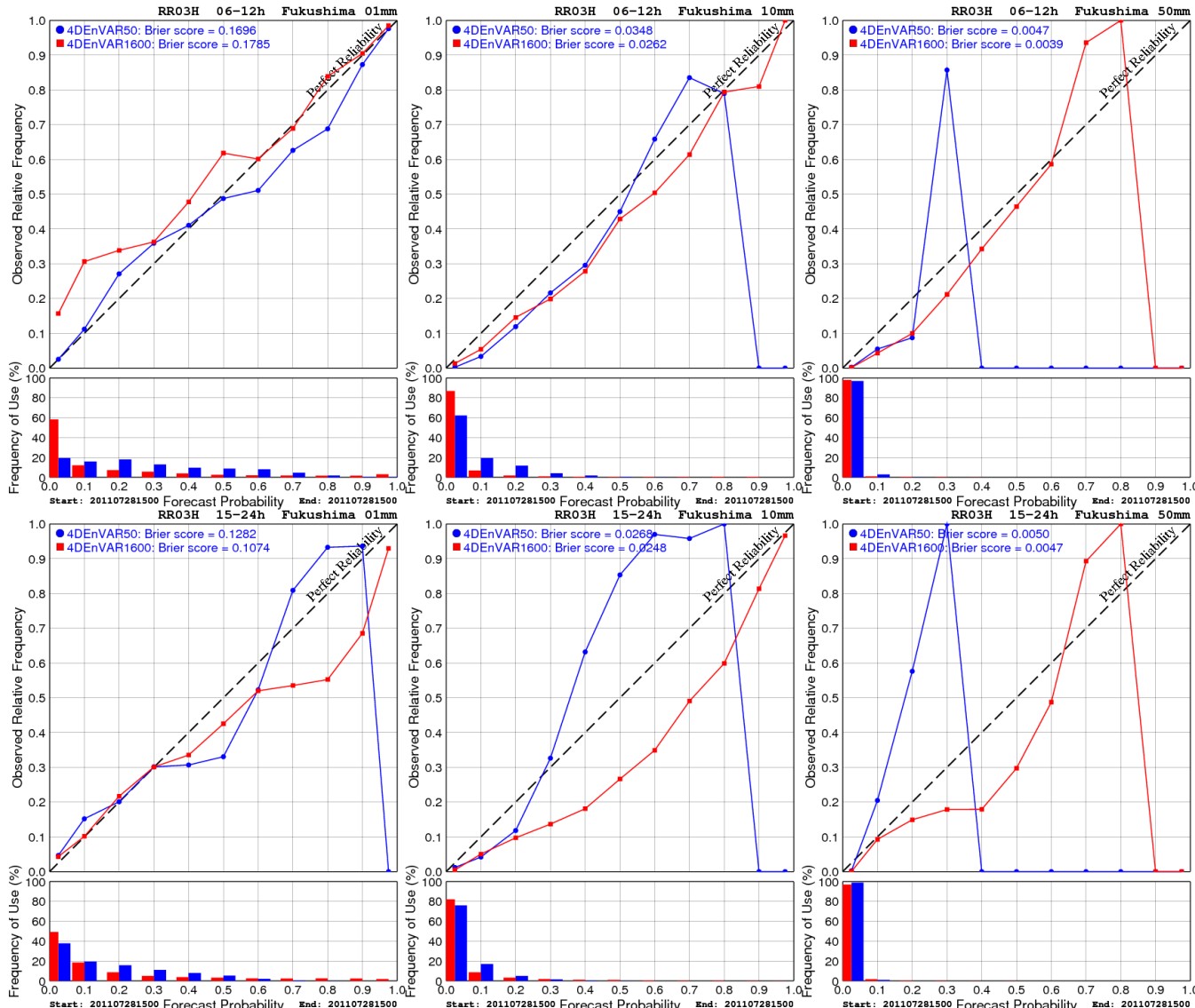

**Figure 2.** As Figure 1 but for reliability diagrams of 3-hour precipitation from ensemble forecasts initialized by analysis ensembles of 4D-EnVAR-NHM using 1600 and 50 members. Three precipitation thresholds of 01 mm (left), 10 mm (center), and 50 mm (right) are chosen. Note that the ensemble forecasts were only run for the time 0000 JST July 29th, 2011.

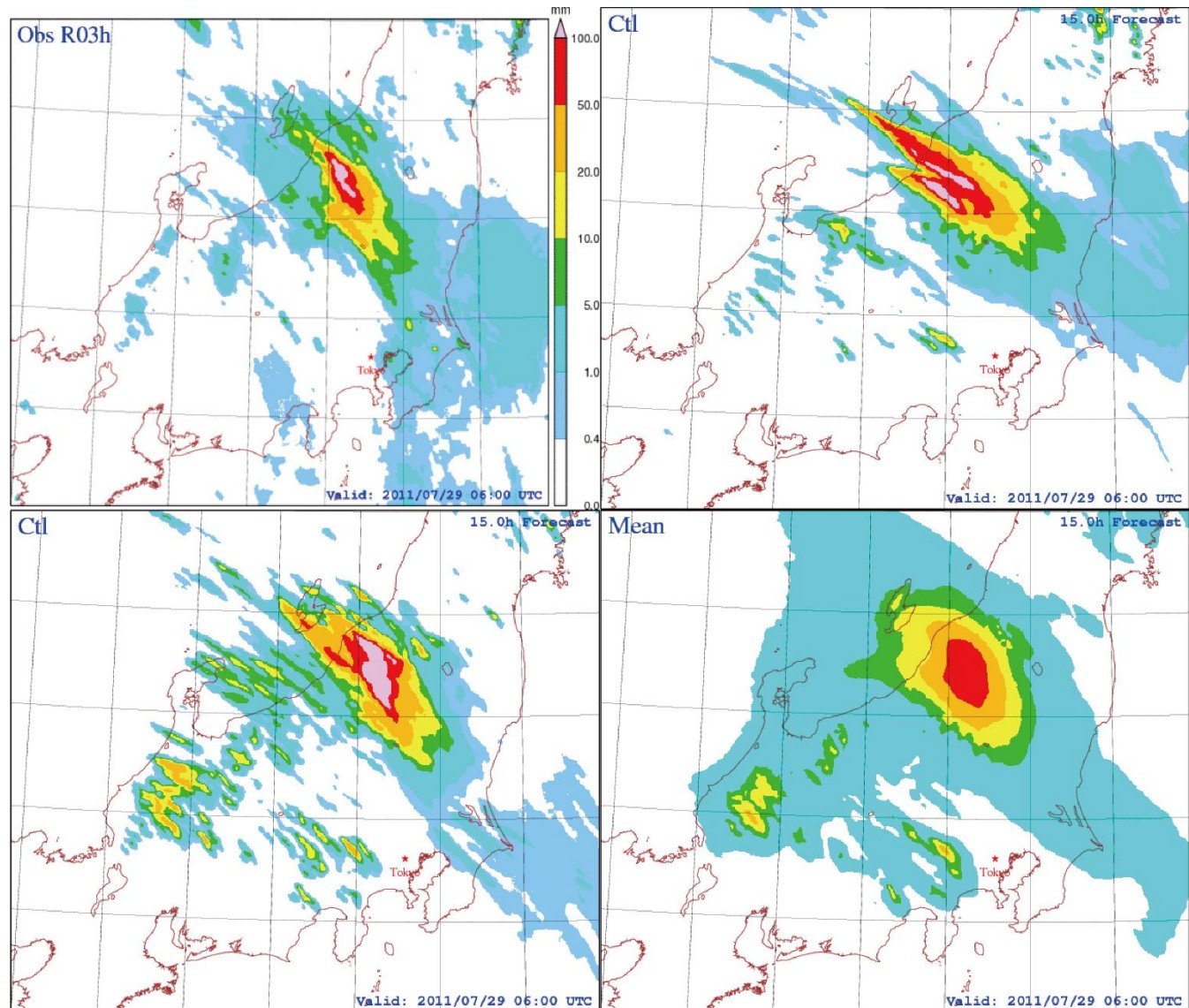

**Figure 3.Three-hour accumulated precipitation for 1200-1500 JST July 29th, 2011 at Fukushima-Niigata as observed by Radar-AMeDAS (RA; top left), forecasted by NHM initialized by the analysis of JNoVA (top right), forecasted by NHM initialized by the analysis of 4D-EnVAR-NHM (bottom left), and the ensemble mean forecast of NHM initialized by the analysis ensemble of 4D-EnVAR-NHM (bottom right). All forecasts were started at 0000 JST July 29th, 2011.**

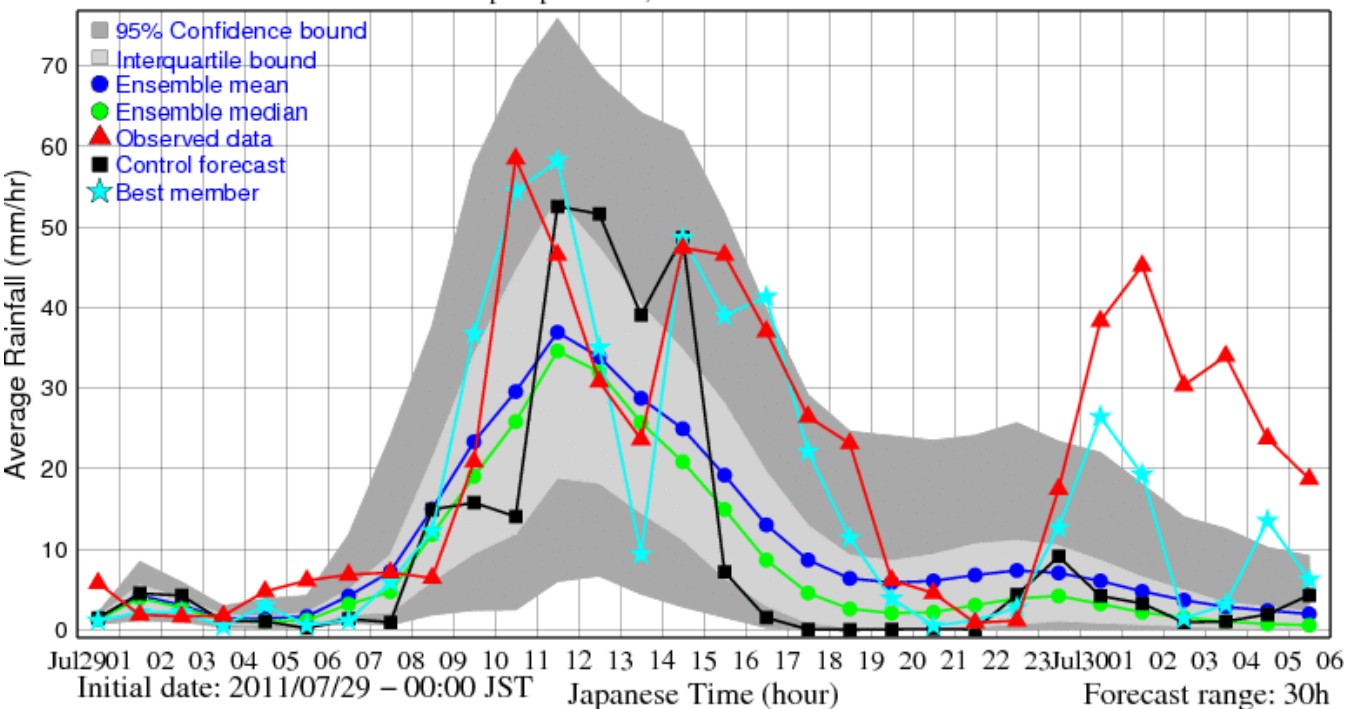

**Figure 4. Time series of one-hour accumulated rainfall over the catchment as forecasted by all ensemble members. The observation, control forecast, ensemble mean forecast, and best member forecast are also plotted for comparison. Here, the best member is defined as the member that has the minimum distance between its time series and the observed time series.**

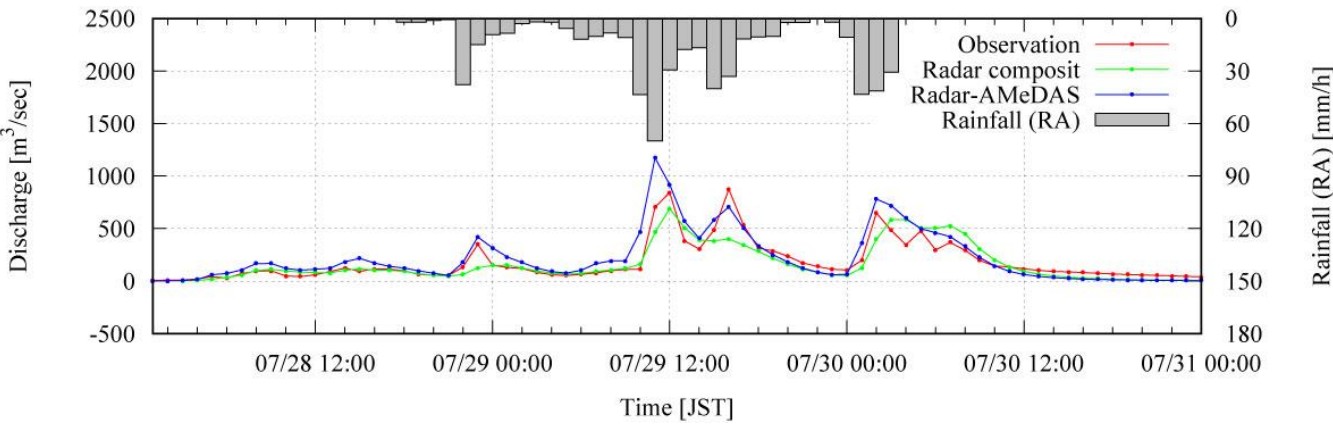

**Figure 5. RA hyetograph, observed dam inflow and simulated inflows with RA and RC.**

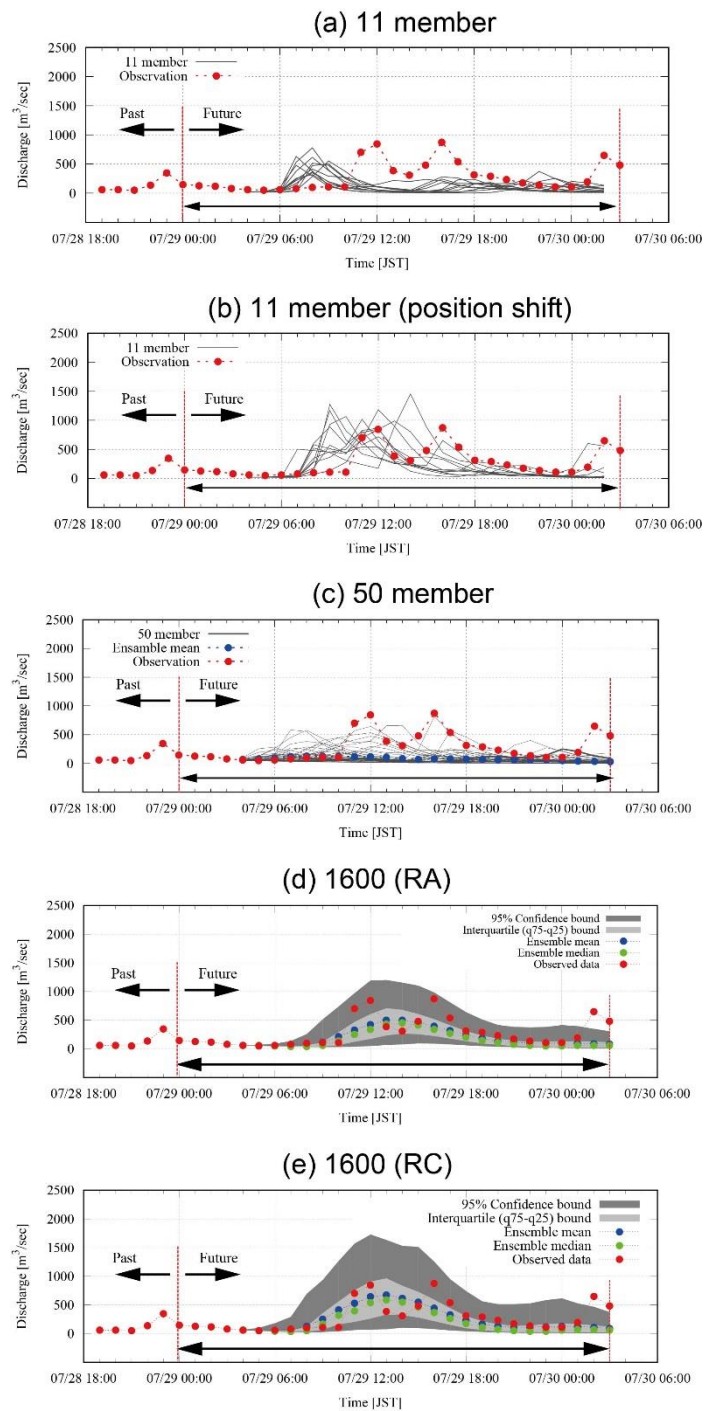

**Figure 6. Hydrographs of (a) 11 discharge simulations in Part 1 (Kobayashi et al., 2016), (b) same 11 member but with a positional shift in Part 1, (c) 50 discharge simulations with 4D-EnVAR-NHM , (d) 1600 discharge simulations with 4D-EnVAR-NHM and model parameters by RA, (e) 1600 discharge simulations with 4D-EnVAR-NHM and model parameters by RC.**

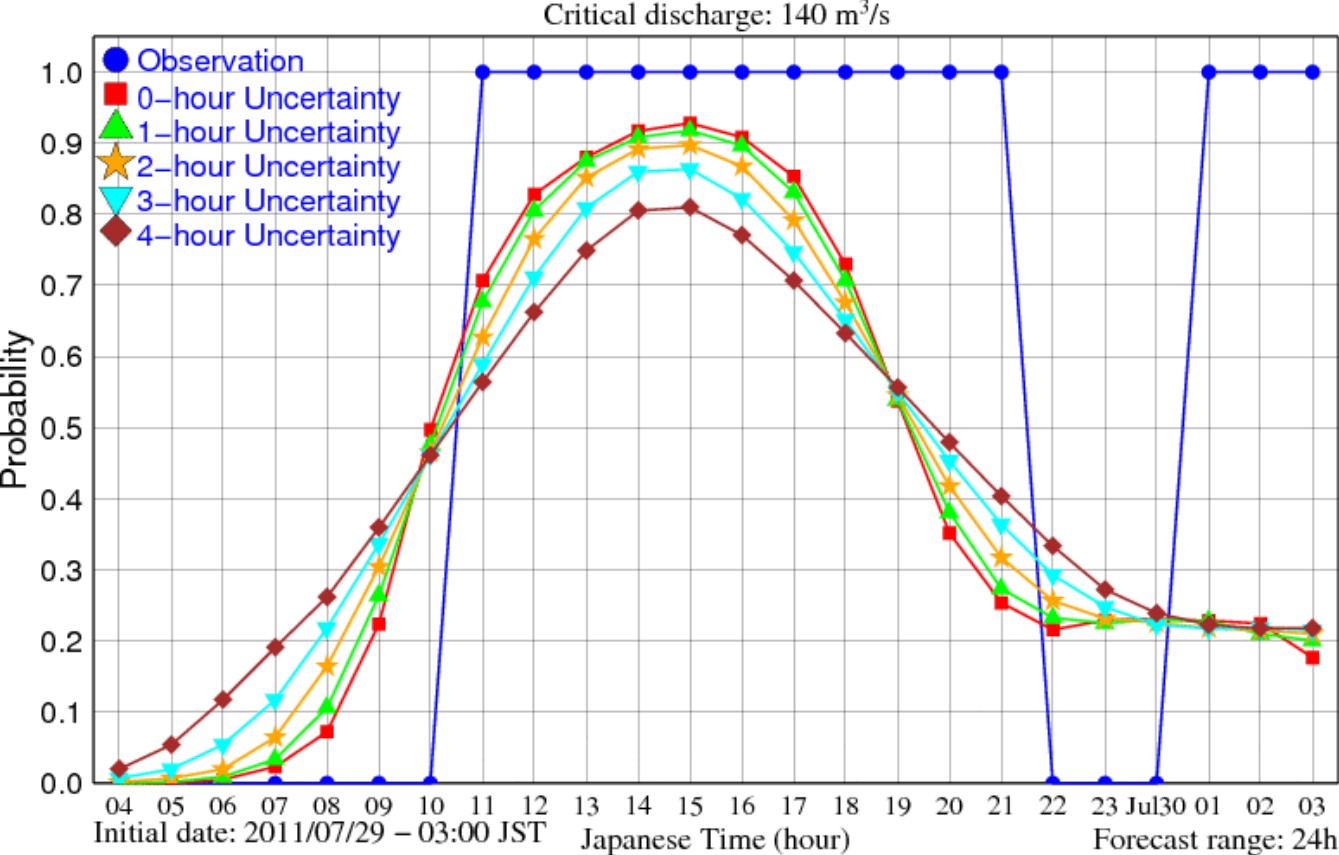

**Figure 7. Probability that the simulated inflow is beyond 140 m³/s considering temporal uncertainty.**

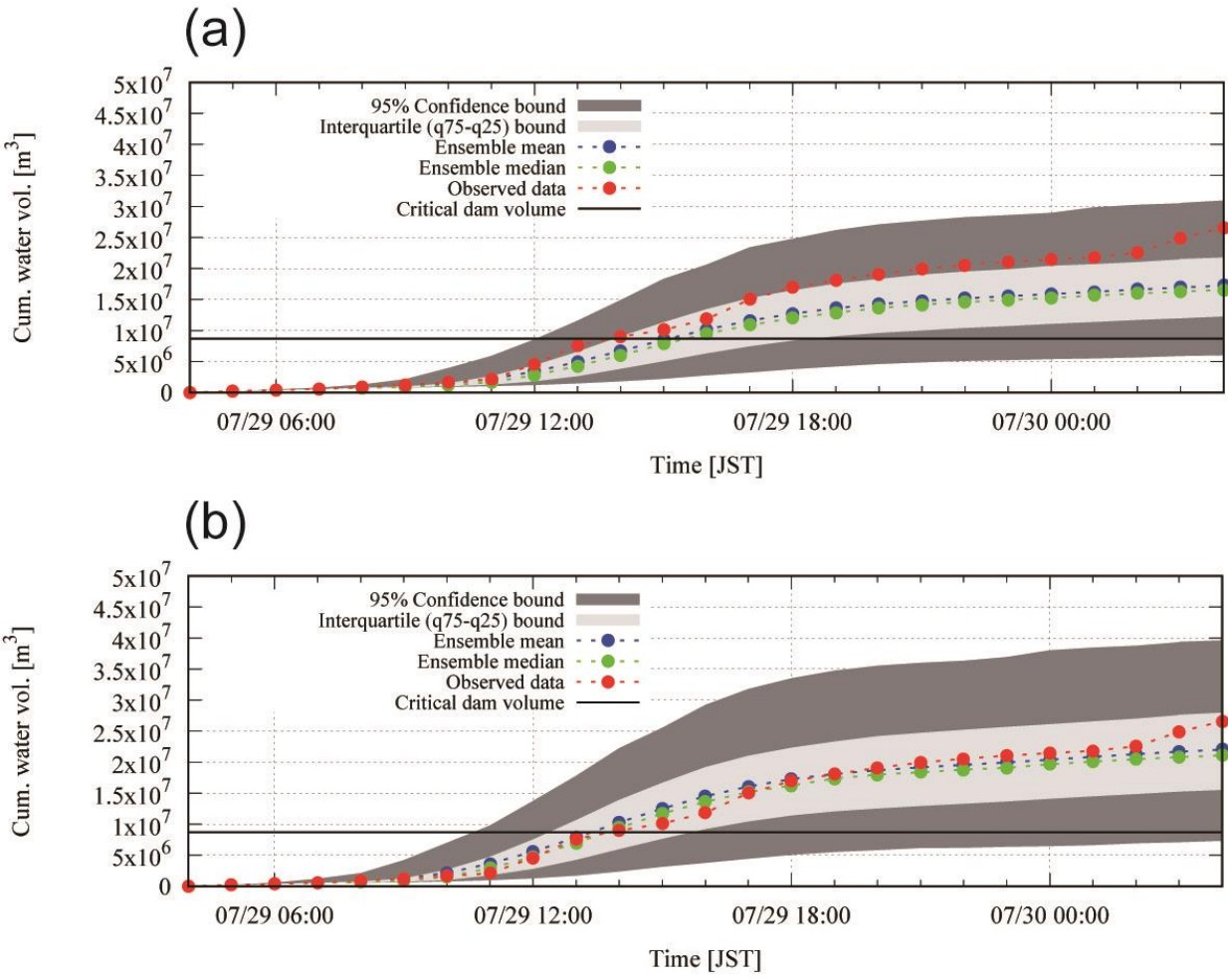

**Figure 8. Cumulative dam inflow by the ensemble simulations, mean of simulation and observations, as well as critical dam volume. (a) with parameters by RA, (b) with parameters by RC.**

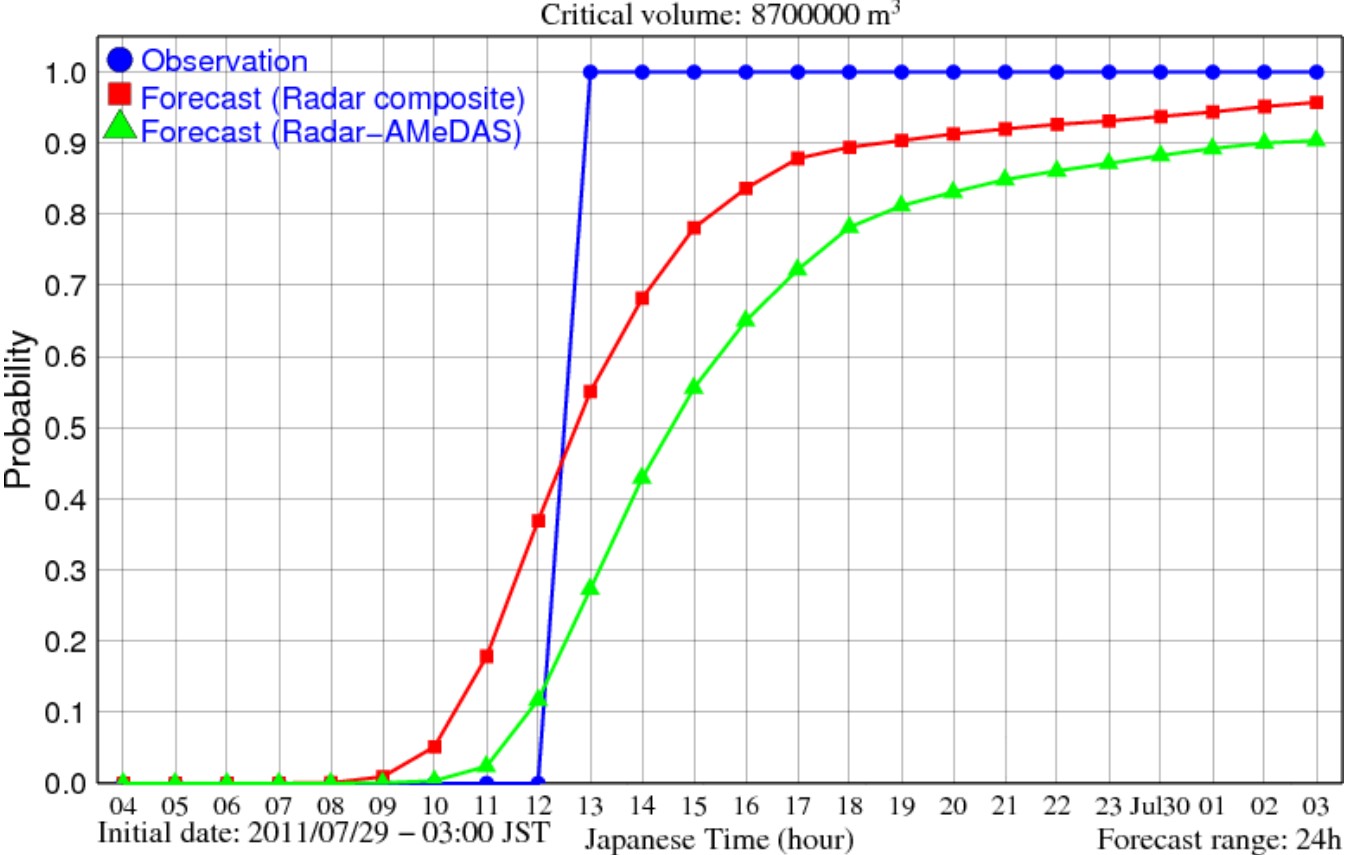

**Figure 9. Probability that the dam needs emergency operation. Radar-AMEDAS and Radar composite indicate the ensemble simulation with the parameters calibrated with the rainfalls.**

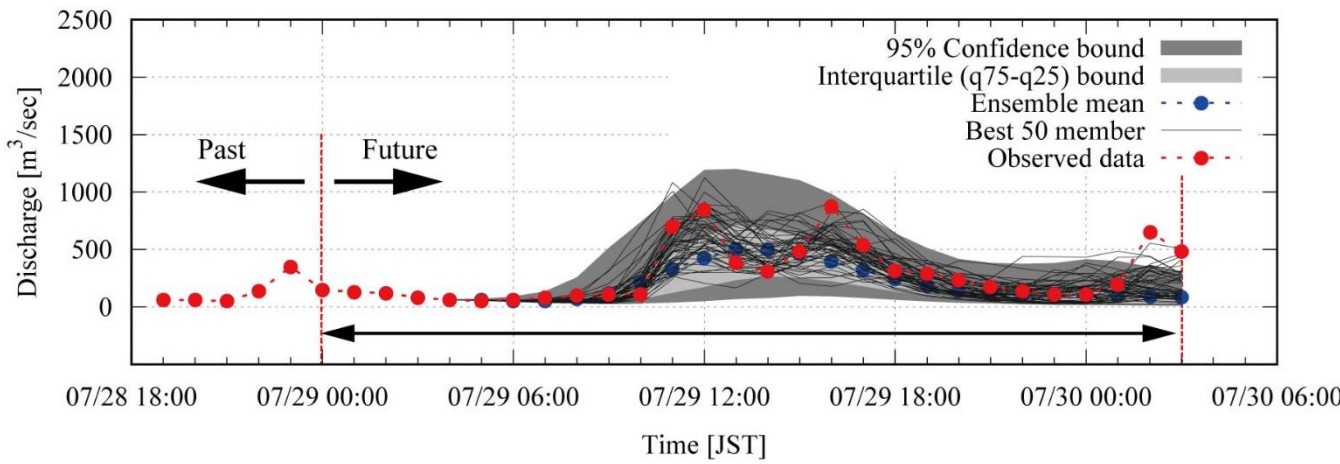

**Figure 10. Hydrographs of all 1600 ensemble members, the 50 best ensemble members (NSE>0.33), and observations.**

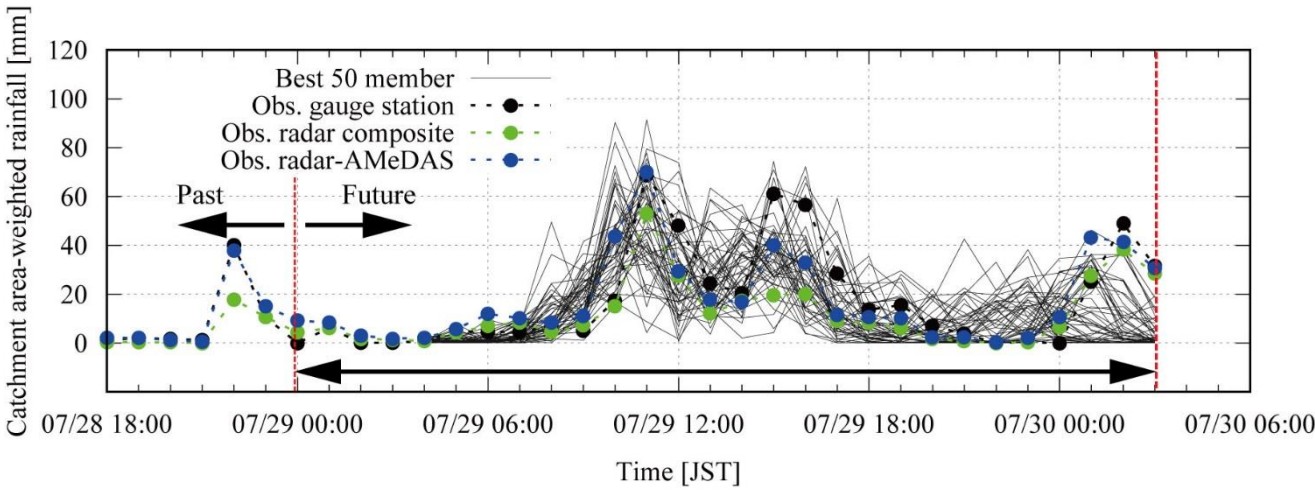

**Figure 11. Rainfall intensity of the 50 best ensemble inflow simulation members, of Radar AMeDAS, of Radar-Composite, and ground observations.**

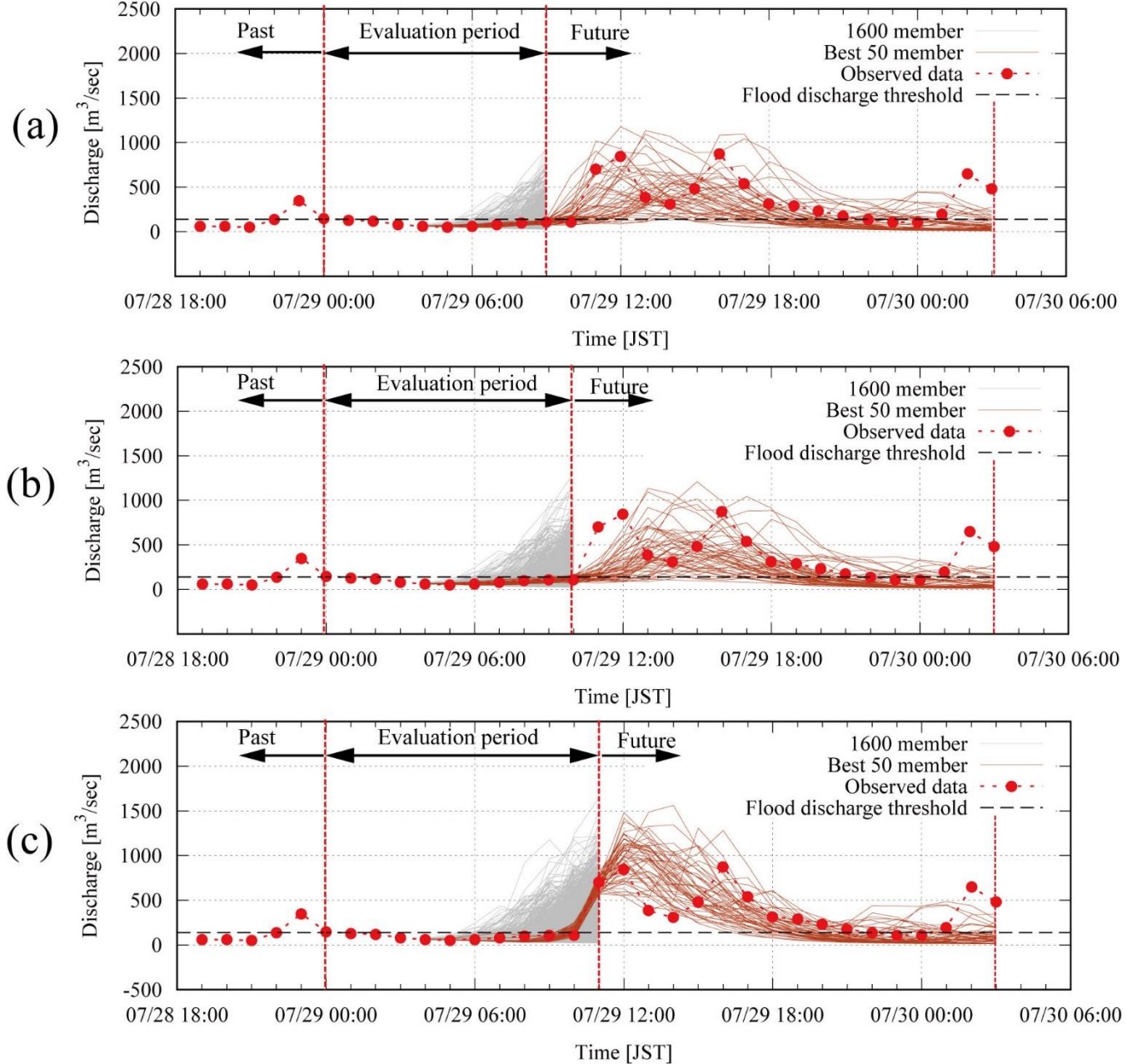

**Figure 12. (a) best 50 ensemble members (NSE >0.24) selected from first 9-hour forecast, (b) best 50 ensemble members (NSE >-0.04) selected from first 10-hour forecast, and (c) best 50 ensemble members (NSE > 0.92) selected from first 11-hour forecast.**

