# Peer review of "Ensemble flood simulation for a small dam catchment in Japan using nonhydrostatic model rainfalls. Part 2: Flood forecasting using 1600 member 4D-EnVAR predicted rainfalls."

_Natural Hazards and Earth System Sciences, 2018_

## Referee Comment (RC1) · Seed (Referee) · 14 Jan 2019

Review of "Ensemble flood simulation for a small dam catchment..." by Kobayashi et al nhess-2018-343

The paper describes the application of a large (1600 members) ensemble of high-resolution rainfall forecasts for flood forecasting in a small (72 km2 ) catchment where the lead time required to respond to a flood warning is longer than the characteristic response time of the catchment. This is an important issue for unban catchments

where hydrological predictions need to be based on rainfall forecasts and not observed rainfall.

The temporal resolution should always be included when discussing resolution. I assume that the ensemble had 10-minute resolution since this is used by the hydrological model.

Section 2 – details of the rainfall event. I looked up the 2016 paper for more details of the meteorological situation, but found very little extra. It would be very helpful to understand better the meteorological situation. I am assuming that, since this case is in Japan and summer, the situation was mostly orographic triggering of severe convection in a very moist airmass. This implies that the model rainfall forecasts are closely forced by the topography where the storms are initiated in the near vicinity of the catchment and are likely to be slow moving? This is important because advection nowcasts will not be able to provide accurate nowcasts in these circumstances.

I really missed some radar rainfall images, say the 10 (or 30)-min rain rates at the times of the three peaks in the hydrograph, just so that we can get a feeling for the space-time structure of the rainfall fields. Actually, the spatial and temporal correlation functions would also be interesting, at least to me as a rainfall person.

Section 6 – Results. It would be very good to extend the results to include a basic analysis of the rainfall forecasts before going to the hydrological verification. In particular, how reliable are the probability of precipitation estimates for the extreme rain rates, especially as a function of ensemble size? This is very important if we are running an ensemble prediction systems to predict the probability of extreme rainfall.

The paper should include some results that show the skill of the model, say the reliability diagram for a high rain rates, as a function of lead time. Did subsequent model runs reproduce the second and third maxima in the hydrograph?

I really liked Figures 7, the probability of the inflow exceeding a critical threshold, and

10, the probability of an emergency operation, as examples of probabilistic products that meets the needs of an end-user. Once again, it would be interesting to see these products for a range of lead times.

Regarding Figure 7, moving the forecasts around in time did not improve the results, but what about moving the ensemble in space? Generally I find that the NWP rainfall forecasts that I work with have limited skill at scales that are below around 100 km. I assume that the rainfall in this case is strongly influenced by the topography so you would not want to shift the rainfall fields too much, but it would still be interesting to move them around by a few tens of km.

The conclusion that it is difficult to select a "set of best ensemble members" based on past performance is significant, if a little discouraging.

Alan Seed

---

## Referee Comment (RC2) · Anonymous Referee #2 · 11 Mar 2019

This manuscript presents the ensemble streamflow forecast forced by a large number of rainfall ensemble at a high resolution. It is a follow-up study of a NHESS paper published in 2016. The main difference is the number of ensemble from 11 to 1600 while the catchment setup is nearly identical. The key idea, use of a large number of rainfall ensemble, is of interest and worth testing since lack of diversity in ensemble has been commonly used as a good excuse for explaining poor performance of hydrologic forecasting. Now, there are 1,600 rainfall ensemble. Can such a large number of rainfall ensemble significantly improve the streamflow forecasting? What factors play

an important role in the improved forecast? These are critical questions hydrological and meteorological communities have been pursuing for a long time. Unfortunately, these questions couldn't be properly answered in the manuscript. For convincing potential readers with new evidence, the experimental setup and analysis methods need significant changes. However, the required changes are too enormous. I have nothing but suggest reject & resubmit. If this manuscript is accepted despite my suggestion, I request the following comments would be addressed before publication.

- Validation at multiple streamflow gauges: Impact of large ensemble forcing should be estimated on multiple gauging locations. If findings are based on the results from a single streamflow gage, feasibility of flood forecasts cannot be claimed while any conclusions can be considered site-specific. In my view, new locations for ensemble verification don't have be limited to dam reservoirs. Any streamflow gages affected by the extreme rainfall are encouraged to be included.

- Selection of a proper size of ensemble: The later part of the result section is about how to find good rainfall ensemble members among 1600 for better streamflow forecasts. The conclusion is vague while all additional efforts are left as future research. Anything additional should be done to draw meaningful findings on this topic. For example, what statistical features do good ensemble have? In addition to rainfall, other meteorological variables may be examined together for analyzing good ensemble. How different or similar ensemble are selected at each time step compared to the previous steps? These questions also should be addressed for multiple gauges, not for a single gauge. If necessary, some machine learning approaches the authors mentioned, e.g. SOM and SVM, may be used in the current manuscript rather remaining them in the future topics.

- To give up the selection of the best ensemble: In the last part of the result section, although it was failed, the authors discussed the possibility of selecting the best discharge simulation using the best rainfall ensemble from 1600. I highly disagree with this idea because ensemble approaches were introduced to overcome the limitation of

deterministic approaches.

- Probabilistic verification: Although this study is about ensemble forecasting, all measures are deterministic, no probabilistic measures are not used for verification of probabilistic forecasts. Since ensemble forecasts aim at providing not only better averages from ensemble but also predictive uncertainty, adequacy of ensemble spread is critical to assessing probability of flooding risks and there are common metrics used in hydrological and meteorological communities for assessing reliability, discrimination, resolution, and sharpness of ensemble. Such metrics should be estimated and discussed.

- Please elaborate why the different number of ensemble was used for each analysis. For figures 11, 13(a), and 13(b), the number of selected ensemble varies from 38 to 26.

- Figure 14. It is negative that any meaningful findings come from simulation results whose NSE values are less than -1. This figure is comparing NSE ranging from 1 to -7.

- Given that the authors also admitted the accuracy of radar rainfall is better than that of NWP, why didn't you use radar rainfall as input for hydrologic modeling in the past time steps? If NWP ensemble are used only for forecasting steps, as most operational models are doing, generally forecast performance is expected to be better.

- Review in Introduction: A simple summary of several papers should be avoided. Previous papers should be used to show how research questions or gaps the current study is dealing with are addressed and remain unsolved.

- The summary of K Project should be removed and, if required, moved to Acknowledgement section because the exascale computing is far from the scope of this journal, despite its importance to the motivation or institutional support to this study.

- Section 4 and several figures on the catchment are nearly identical to Section 3 and

associated figures in the 2016 NHESS paper, which should be considered as self-plagiarism if not cited properly.

---

## Author Response (AR1)

**Authors' response**

We would like to thank the editor and two reviewers for the detailed and thoughtful comments and the time that you spent on the article. All your invaluable comments have inspired us to rethink carefully the problem that we tried to investigate. With our revision we hope that you will find it a better article.

Regarding the manuscript with track changes, we did not use the MS-word track change function since the changes are really a lot so that the manuscript became hard to be understood with the complex track change indications. Instead, we highlighted the changes by ourselves using different colors. If this is a problem, please tell us.

In addition, we have modified the abstract according to the comments from the editor and two reviewers to show more clearly about the object of our Part 2 paper. We tried to answer the rest concerns from the editor in the responses below. We hope it can satisfy you.

Finally, we would like to change the order of the authors as indicated in the revised manuscript. We hope that you can accept the changes.

**Reviewer 1:**

(1) The paper describes the application of a large (1600 members) ensemble of high-resolution rainfall forecasts for flood forecasting in a small (72 km$^2$) catchment where the lead time required to respond to a flood warning is longer than the characteristic response time of the catchment. This is an important issue for unban catchments where hydrological predictions need to be based on rainfall forecasts and not observed rainfall.The temporal resolution should always be included when discussing resolution. I assume that the ensemble had 10-minute resolution since this is used by the hydrological model.

Reply: Since 4D-EnVAR-NHM outputted data every hour due to limited data storage, we also applied the hourly data to the rainfall-runoff model.

(2) Section 2 – details of the rainfall event. I looked up the 2016 paper for more details of the meteorological situation, but found very little extra. It would be very helpful to understand better the meteorological situation. I am assuming that, since this case is in Japan and summer, the situation was mostly orographic triggering of severe convection in a very moist airmass. This implies that the model rainfall forecasts are closely forced by the topography where the storms are initiated in the near vicinity of the catchment and are likely to be slow moving? This is important because advection nowcasts will not be able to provide accurate nowcasts in these circumstances.

Reply: We have added a paragraph in the revised manuscript to analyze the meteorological situation in more details, which we reproduce here:

For the details of the 2011 Niigata-Fukushima heavy rainfall, see our Part 1 paper (Kobayashi et al., 2016). An additional note is that the torrential rain of the 2011 Niigata-Fukushima heavy rainfall occurred over the small area along the synoptic scale stationary front (for surface weather map, see Fig. 1 of Kobayashi et al. 2016). Saito et al (2013) conducted two 11-member downscale ensemble forecasts with different horizontal resolutions (10 and 2 km) for this event using JMA-NHM and JMA's global ensemble EPS perturbations. They found that the location where intense rain concentrates variable to small changes of model setting, thus the position of the heavy rain was likely controlled mainly by horizontal convergence along the front, rather than the orographic forcing.

(3) I really missed some radar rainfall images, say the 10 (or 30)-min rain rates at the times of the three peaks in the hydrograph, just so that we can get a feeling for the space-time structure of the rainfall fields. Actually, the spatial and temporal correlation functions would also be interesting, at least to me as a rainfall person.

Reply: For supplement information, we would like to show here the radar images corresponding to the times when the three peaks occurred in the hydrograph (Figure S1). We do not intend to add this figure into the manuscript.

[Figure]

Figure S1. Reflectivity from radar composition at the time of the first (left), second (center), and third (right) peaks of the hydro-graph.

(4) Section 6 – Results. It would be very good to extend the results to include a basic analysis of the rainfall forecasts before going to the hydrological verification. In particular, how reliable are the probability of precipitation estimates for the extreme rain rates, especially as a function of ensemble size? This is very important if we are running an ensemble prediction system to predict the probability of extreme rainfall. The paper should include some results that show the skill of

the model, say the reliability diagram for high rain rates, as a function of lead time. Did subsequent model runs reproduce the second and third maxima in the hydrograph?

Reply: We have added two paragraphs to Section 3 describing the verification results of the rainfall forecasts. We agree with the reviewer's comment that verification scores for rainfall forecasts with respect to different lead times should be included in the paper. However, it was very costly to run a high-resolution (2 km grid spacing) ensemble forecast using 1600 members even for a specific time. Due to this reason, we could only run deterministic forecasts for all other initial times and use the Fraction Skill Score to measure forecast performance of the deterministic forecasts at different lead times. We had also run an additional experiment using only 50 ensemble members to compare with the case using 1600 members. Reliability diagrams are then plotted for these ensemble forecasts by the two experiments, even though we only run the ensemble forecasts at a specific time. Of course, the FSSs are also calculate for this additional experiment. Here are the paragraphs that we have added to the revised manuscript:

"Due to limited computational resource, ensemble forecasts with 1600 members were only employed for the target time of 0000 JST July 29th, 2011. However, deterministic forecasts were run for all other initial times to examine impact of number of ensemble members on analyses and the resulting forecasts. Figure 1 shows the verification results for the 3-hour precipitation forecasts as measured by the Fraction Skill Score (FSS) (Duc et al., 2013). Here we aggregate the 3-hour precipitation in the first and second 12-hour forecasts to increase samples in calculating the FSS. By this way, robust statistics are obtained but at the same time dependence of the FSS on the leading times can still be shown. Note that an additional experiment with 4D-EnVAR-NHM using 50 ensemble members, which is called 4DEnVAR50 to differentiate with the original one 4DEnVAR1600, was run. It is very clear from Figure 1 that 4DEnVAR1600 outperforms 4DEnVAR50 almost for all precipitation thresholds, especially for intense rain. Also for high rain-rate, compared to JNoVA, 4DEnVAR1600 forecasts are worse than JNoVA forecasts for the first 12-hour forecasts, which can be attributed to the fact that 4D-EnVAR-NHM did not assimilate satellite radiances and surface precipitation like JNoVA. However, it is interesting to see that 4D-EnVAR-NHM produces forecasts better than JNoVA for very intense rains for the next 12-hour forecasts.

To check reliability of the ensemble forecasts, reliability diagrams are calculated and plotted in Figure 2 for 4DEnVAR1600 and 4DEnVAR50. Since JNoVA only provided deterministic forecasts, reliability diagram is irrelevant for JNoVA. Note that we only performed ensemble forecasts initialized at the target time of 0000 JST July 29th, 2001 due to lack of computational resource to run 1600-member ensemble forecasts at different initial times. Therefore, the same strategy of aggregating 3-hour precipitation over the first and second 12-hour forecasts in calculating the FSS in Figure 1 is applied to obtain significant statistics. Clearly, Figure 2 shows that 4DEnVAR1600 is distinctively more reliable than 4DEnVAR50 in predicting intense rain. While 4DEnVAR50 cannot capture intense rain, 4DEnVAR1600 tends to overestimate areas of intense rain. The tendency of

overestimation of 4DEnVAR1600 becomes clearer if we consider the forecast ranges between 12 and 24 hours. However, for the first 12 hours, 4DEnVAR1600 slightly underestimates areas of light rains. This also explains why the FSSs of 4DEnVAR1600 are smaller than those of 4DEnVAR50 for small rainfall thresholds in Figure 1."

Since we only run deterministic forecasts for other initial times, we show here the forecast results for other lead times as supplement information (Figure S2). Again, we do not intend to add this figure into the manuscript. It turns out that it is more difficult to forecast the second and third peaks in the hydrograph.

[Figure]

Figure S2. Time series of one-hour accumulated rainfall over the catchment by deterministic forecasts of JNoVA (top) and 4DEnVAR1600 (bottom) at different lead times.

(5) I really liked Figures 7, the probability of the inflow exceeding a critical threshold, and 10, the probability of an emergency operation, as examples of probabilistic products that meets the needs of an end-user. Once again, it would be interesting to see these products for a range of lead times. Regarding Figure 7, moving the forecasts around in time did not improve the results, but what about moving the ensemble in space? Generally, I find that the NWP rainfall forecasts that I work with have limited skill at scales that are below around 100 km. I assume that the rainfall in this

case is strongly influenced by the topography so you would not want to shift the rainfall fields too much, but it would still be interesting to move them around by a few tens of km.

Reply: As explained above, our computational resource can only afford running 1600-member ensemble forecast for a specific time. Although it's desirable to know the forecasts as plotted Figure 7 for other lead times, limited computation resource prevented us to employ this. In design the plot in Figure 7, we introduced the idea of using spatial and temporal uncertainty in verification from the FSS into the hourly discharges. It is clear that hourly discharges have strong correlation with hourly precipitation. Then it is reasonable to consider temporal uncertainty in hourly precipitation. Since the rainfall over the catchment here is not rainfall at any specific grid point but rainfall over many grid points (more than 70 $km^2$ in our problem). Therefore, temporal uncertainty is more relevant to hourly catchment rainfall rather than spatial uncertainty. Also, computation with spatial uncertainty is more complicated in this case since we must consider all directions of displacement vectors in a two-dimensional space, which have more degree of freedom that just one direction in the one-dimensional space of temporal uncertainty. Therefore, we do not consider spatial uncertainty in plotting Figure 7 and 10.

Nevertheless, please kindly note that the supplemental information for the spatial uncertainty was then written to reply the reviewer 2 (1) comment.

(6) The conclusion that it is difficult to select a "set of best ensemble members" based on past performance is significant, if a little discouraging.

Reply: The second reviewer has also shown interest on this problem. In the revised version, we have tried to established a theoretical framework to support the idea of "the best ensemble members". From this theory, we have explained why the best ensemble members based on past performance can vary considerably in time and this makes selection of best ensemble members difficult.

**Reviewer 2:**

(1) This manuscript presents the ensemble streamflow forecast forced by a large number of rainfall ensemble at a high resolution. It is a follow-up study of a NHESS paper published in 2016. The main difference is the number of ensemble from 11 to 1600 while the catchment setup is nearly identical. The key idea, use of a large number of rainfall ensemble, is of interest and worth testing since lack of diversity in ensemble has been commonly used as a good excuse for explaining poor performance of hydrologic forecasting. Now, there are 1,600 rainfall ensemble. Can such a large number of rainfall ensemble significantly improve the streamflow forecasting? What factors play an important role in the improved forecast? These are critical questions hydrological and meteorological communities have been pursuing for a long time.

Unfortunately, these questions couldn't be properly answered in the manuscript. For convincing potential readers with new evidence, the experimental setup and analysis methods need significant changes. However, the required changes are too enormous. I have nothing but suggest reject & resubmit. If this manuscript is accepted despite my suggestion, I request the following comments would be addressed before publication.

Reply: First, we agree with the reviewer that impact of large weather ensemble forcing on hydrological forecasts are worth to examine in details. However, this interesting research topic itself is not the main topic that we would like to pursue this time in this paper as the Part 2 in a series on the hydrological forecast for the Kasahori dam. Our purpose in using the 1600-member ensemble forecast was only that we have found the rainfall forecast for the large area covering the small catchment of the Kasahori dam was significantly improved when using this ensemble forecast and we have hoped that this helps, as a result, to improve the streamflow forecast for the Kasahori dam. The impression that the topic of our research is on impact of large ensemble forcing on hydrological forecasts will not occur if, instead of the 1600-member ensemble forecast, we used a 50-member ensemble forecast in the paper. In fact, we have tried to use a 50-member ensemble forecast with the same data assimilation system 4D-EnVAR-NHM, however the performance of the rainfall forecast was not better than the operational forecast that we used in Part 1. Of course, the improvement rooted in the use of very large ensemble members and this problem involved to the way that this large ensemble members mitigate site-effects of localization in 4D-EnVAR-NHM by removing vertical localization and imposing a unique horizontal localization length scale for all variables at all vertical levels. We have described these special characteristics of 4D-EnVAR-NHM in the text.

Likewise, as the information from hydrological aspects, we would like to show Figures S3 and S4 below (Figures 6 and 7 in the revised manuscript this time). Figure S3 shows the comparisons of the hydrographs of (a) 11 discharge simulations in Part 1 (cited from Kobayashi et al., 2016), (b) same 11 member but with a positional shift in Part 1 (Kobayashi et. al. 2016), (c) 50 discharge simulations with 4D-EnVAR-NHM and (d) 1600 discharge simulations with 4D-EnVAR-NHM. In addition, Figure S4 shows the histogram of NSE based on the simulated and observed discharges for the 4 cases.

First, Figure S4 (c) and (d) are showing the large improvement of 1600 ensemble discharge simulation in terms of NSE compared with 50 ensemble simulations. Second, Figure S4 (a) and (c) are showing that the performance of the discharge forecast with 50 member 4D-EnVAR-NHM was not necessarily better than the discharge forecast with the forecast in Part 1 in terms of NSE. Likewise, Figure S4 (b) shows that the discharge forecasts could be improved by positional shift of the rainfall field in Part 1, though this positional shift still needs further statistical verification with more rainfall events.

With these analysis, the explanation below is added in pages 8 and 9 of the manuscript.

"Figure 6 shows the comparisons of the hydrographs of (a) 11 discharge simulations in Part 1, (b) same 11 member but with a positional shift in Part 1, (c) 50 discharge simulations with 4D-EnVAR-NHM and (d) 1600 discharge simulations with 4D-EnVAR-NHM. Note that the duration of the 4D-EnVAR-NHM ensemble weather simulation is 30 hours from 0000 July 29th to 0700 July 30th JST, but the ensemble flood simulation is carried out only for 24 hours from 0300 July 29th to 0300 July 30th, 2011 JST since we consider that JMA-NHM uses the first 3 hours to adjust its dynamics. The result in Figure 6 (d) shows that, except for the third peak, the 1600 ensemble inflows can encompass the observed rainfall within the range, which was not realized in Part 1 with 11 downscale ensemble rainfalls of 2 km resolution (Figure 6 (a)). In other words, the extreme rainfall intensity of the event can be reproduced by the ensemble members with 1600 4D-EnVAR-NHM. Likewise, comparing Figure 6 (c) and (d), the simulated discharges by 50 ensemble rainfalls of 4D-EnVAR-NHM encompass the observation within the range less than those of 1600 ensemble members. In addition, Figure 7 shows the histogram of NSE based on the simulated and observed discharges for the 4 cases. Looking at Figure 7 (c) and (d), clearly, the 1600 ensemble discharge simulations outperform 50 ensemble simulations. The NSE>0 was around 17.75% (284 members) in 1600 ensembles, while it was 0% in 50 ensembles. On the other hand, Figures 7 (a) and (c) show that the performance of the discharge forecast with 50 member 4D-EnVAR-NHM was not necessarily better than the 11 discharge forecast in Part 1 in terms of NSE. Likewise, Figures 6 (b) and 7 (b) show that the discharge forecasts could be improved by the positional shift of the rainfall field in Part 1, though this positional shift still needs further statistical verification with more rainfall events."

[Figure]

Figure S3. Hydrographs of (a) 11 discharge simulations in Part 1 (Kobayashi et al., 2016), (b) same 11 member but with a positional shift in Part 1, (c) 50 discharge simulations with 4D-EnVAR-NHM and (d) 1600 discharge simulations with 4D-EnVAR-NHM.

[Figure]

Figure S4. Histograms of NSE based on the simulated and observed discharges to Kasahori dam: (a) 11 discharge simulations in Part 1 (Kobayashi et al., 2016), (b) same 11 member but with a positional shift  in Part 1 (Kobayashi et. al. 2016), (c) 50 discharge simulations with 4D-EnVAR-NHM and (d) 1600 discharge simulations with 4D-EnVAR-NHM.

(2) - Validation at multiple streamflow gauges: Impact of large ensemble forcing should be estimated on multiple gauging locations. If findings are based on the results from a single streamflow gauge, feasibility of flood forecasts cannot be claimed while any conclusions can be considered site-specific. In my view, new locations for ensemble verification don't have be limited to dam reservoirs. Any streamflow gages affected by the extreme rainfall are encouraged to be included.

Reply: As we have explained in the previous answer, our research focused on the hydrological forecast for the Kasahori dam but not on impact of large ensemble forcing. Thus, it is inappropriate here to verify the hydrological ensemble forecast at all gauging locations in the domain covered by the meteorological forecast. We agree with the reviewer that if the topic is on impact of large ensemble forcing, conclusion should be relied on verification at all gauging locations. However, in this case verification for rainfall forecast over the whole domain is enough to draw conclusion on

performance of the ensemble forecast. We have already provided this kind of verification in Section 3 of the manuscript.

(3) - Selection of a proper size of ensemble: The later part of the result section is about how to find good rainfall ensemble members among 1600 for better streamflow forecasts. The conclusion is vague while all additional efforts are left as future research. Anything additional should be done to draw meaningful findings on this topic. For example, what statistical features do good ensemble have? In addition to rainfall, other meteorological variables may be examined together for analyzing good ensemble. How different or similar ensemble are selected at each time step compared to the previous steps? These questions also should be addressed for multiple gauges, not for a single gauge.

Reply: We agree with the reviewer that this section lacks a rigorous theory on guiding to choose the best ensemble members that supports our conclusions. We have added this theory on the revised manuscript that we would like to copy here:

[revised manuscript text omitted]

The theory can be traced back to the theory of data assimilation in which the mathematical form of particle filter can be served as a foundation for choosing the best ensemble members. Relied on available observations, a weight can be assigned for each ensemble members, which in fact represents the likelihood of the observations conditioned on this forecast. This reduces to verification scores for each member depending on the form of the likelihood. We would like to remark here that the theory is applied for any number of ensemble members.

Thus, statistical features that define the best ensemble members are the relative likelihood $w_i^{post} = \frac{p_Y(\mathbf{y}|\mathbf{x}_i)}{\sum_{j=1}^{K} p_Y(\mathbf{y}|\mathbf{x}_j)}$. These likelihoods vary with the lead time, therefore at different time steps, we will have different good ensemble members. The observation $\mathbf{y}$ here denotes all kinds of observations that means the relative likelihoods encompass not only rainfall but also other meteorological variables available at the first few hours. However, if we have already known the meteorological observations, it is better to run the next assimilation cycle, and using the new analysis ensemble to produce the new ensemble forecast. It is more practical to assume that we only know observations of discharge and rainfall at the first few hours.

(4) - If necessary, some machine learning approaches the authors mentioned, e.g. SOM and SVM, may be used in the current manuscript rather remaining them in the future topics.

Reply: We have removed the call for machine learning methods in the manuscript since this is not relevant now. On the other hand, we have added the text below to the page 13 of the revised manuscript.

"Herein lies the problem that, NSEs are quite sensitive to spatial and temporal displacement errors in rainfall. In principle, it is possible to introduce those errors into NSEs in a way similar to FSSs. However, it should be cautious in introducing such errors into NSEs before investigated well, although this type of approach has been used frequently in meteorology community. How to incorporate them qualitatively is also a problem to be addressed."

We have tested to incorporate the temporal displacement errors of rainfall into discharge NSE. One result is shown in Figure S5 below. The figure shows NSE considering temporal shift of ±1, 2 hr. It shows clearly that the number of discharge simulation members with NSE >0 increases, e.g. if we consider time-lag of +1hr. Nevertheless, as this analysis needs more elaboration, we would like to show them temporally as your information in this reply letter only.

[Figure]

Figure S5. NSE considering temporal displacement errors.

(5) - To give up the selection of the best ensemble: In the last part of the result section, although it was failed, the authors discussed the possibility of selecting the best discharge simulation using the best rainfall ensemble from 1600. I highly disagree with this idea because ensemble approaches were introduced to overcome the limitation of deterministic approaches.

Reply: When we select the best ensemble members based on the observations at the first few hours, we have already replaced the original ensemble forecast given by the sample ($\mathbf{x}_i$, $w_i^{pre} = 1/K$, i=1,..,K) by the new sample consisting of a small number of the best ensemble members. The old manuscript did not describe the form of this resulting pdf, and mistakenly assumed that the new ensemble forecast is populated from an equally weighted sample ($\mathbf{x}_j^{best}$, $w_j^{best} = 1/N$, j=1,..,N). Furthermore, we lost useful information when moving from a large ensemble to a small ensemble. And, we have somehow agreed with the point of view of the reviewer on this problem.

Under the theory on selection of the best ensemble members that we have introduced in the revised version, the predictive pdf yielded by the best ensemble members is now given its explicit form in (6). The theory shows that the notion of the best ensemble members still has a certain application if many members have very small relative likelihood $w_i^{post} = \frac{p_Y(\mathbf{y}|\mathbf{x}_i)}{\Sigma_{j=1}^K p_Y(\mathbf{y}|\mathbf{x}_j)}$, and the predictive pdf is dominated by a small number of ensemble members. The weights $w_i^{post}$ strongly depend on our

model on observation errors.

(6) - Probabilistic verification: Although this study is about ensemble forecasting, all measures are deterministic, no probabilistic measures are not used for verification of probabilistic forecasts. Since ensemble forecasts aim at providing not only better averages from ensemble but also predictive uncertainty, adequacy of ensemble spread is critical to assessing probability of flooding risks and there are common metrics used in hydrological and meteorological communities for assessing reliability, discrimination, resolution, and sharpness of ensemble. Such metrics should be estimated and discussed.

Reply: The first reviewer has also shown the same concern on rainfall verification. In the revised manuscript we have added reliability diagrams for rainfall forecasts. Ensemble spreads have in deed given in the plots for 1600 forecasts of the rainfall and discharge at the catchment (Figures 2 and 6 in the original manuscript), but under the form of inter-quartiles in the box-and-whisker diagrams. The use of inter-quartile is more robust than the normal spread since the latter is more sensitive to outliers. Another probabilistic score (the Brier score) has been shown implicitly in Figures 7 and 10 (in the original manuscript) for discharge and accumulated volume when we predefined some thresholds. Instead of plotting all forecast probabilities and the corresponding observations, the Brier scores can be computed from all lead times. However, it is more informative to plot all pairs of the forecast probabilities and the observations.

(7) - Please elaborate why the different number of ensemble was used for each analysis. For figures 11, 13(a), and 13(b), the number of selected ensemble varies from 38 to 26.

Reply: The number (i.e. 26 to 38) corresponds to the number of (NSE>0, NSE>0.25, NSE>0.9 etc.). Thus, it is not same each other. But we decided to make the number all 50 and replotted the figures considering your comments

(8) - Figure 14. It is negative that any meaningful findings come from simulation results whose NSE values are less than -1. This figure is comparing NSE ranging from 1 to -7.

Reply: We have removed this figure in the revised manuscript since the theory on selection of the best members introduced in the revised manuscript makes this figure irrelevant now.

(9) - Given that the authors also admitted the accuracy of radar rainfall is better than that of NWP, why didn't you use radar rainfall as input for hydrologic modeling in the past time steps? If NWP ensemble are used only for forecasting steps, as most operational models are doing, generally forecast performance is expected to be better.

Reply: Yes, we can use radar rainfall available at the first few hours to run our hydrological model.

However, in this case, we can no longer select the best members from forecasts for the first few hours. In this study one of the topics is to know whether we can infer the best members from the observations at the first few hours. If the answer is negative, we agree that your approach is the most appropriate way to improve forecast performance. But of course, in this way we need to run all 1600 members.

(10)     - Review in Introduction: A simple summary of several papers should be avoided. Previous papers should be used to show how research questions or gaps the current study is dealing with are addressed and remain unsolved.

Reply: We have added several studies focusing on rainfall forecasts at cloud-resolving scales around mountainous areas, to show the importance of cloud-resolving ensemble weather simulation especially as the input to the rainfall-runoff model. Those are written in the revised manuscript as follows:

"Likewise in Europe, Hohenegger (2008) carried out the cloud-resolving ensemble weather simulations of the August 2005 Alpine flood. Their cloud resolving EPS of 2.2 km grid space included the explicit treatment of deep convection and was the result of downscaling of COSMO-LEPS (10km resolution driven by ECMWF EPS).   Their conclusion was that despite the overall small differences, the 2.2 km cloud resolving ensemble produces results as good as and even better than its 10km EPS, though the paper did not deal with the hydrological forecasting. Another paper which dealt with cloud resolving ensemble simulations can be found in Vie et al. (2011) for Mediterranean heavy precipitation event. Their ensemble weather simulation model resolution was 2.5 km by AROME from Meteo-France which uses ALADIN forecast for lateral boundary condition (10km resolution), thus the deep convection was explicitly resolved. We can recognize from these researches that the European researchers especially around mountain region have been farsighted from early days for the importance of these cloud resolving ensemble simulations."

Likewise, we have also added some clear explanation about the scope of our papers Part 1 and 2 in series. Those are written in the revised manuscript as follows.

"Since the new EPS produced better forecasts of the rainfall field, in this study, as a Part 2 version of Kobayashi et al. (2016), we applied those 1600 ensemble rainfalls to the ensemble inflow simulations to Kasahori Dam without the positional lag correction. The main theme of this Part 2 paper is that the 1600 ensemble rainfall forecasts can significantly improve the rainfall forecast over the large area around Kasahori dam and this would, as a result, help to improve the streamflow forecast for the Kasahori dam. In the series of Part 1 and 2, we intentionally have chosen a rainfall-runoff model whose specification is quite close to those runoff models used in many governmental practices of Japanese flood forecasting to see the usefulness of 1600 ensemble rainfalls."

(11)     - The summary of K Project should be removed and, if required, moved to Acknowledgement section because the exascale computing is far from the scope of this journal, despite its importance to the motivation or institutional support to this study.

Reply: These sentences are removed.

(12)     - Section 4 and several figures on the catchment are nearly identical to Section 3 and associated figures in the 2016 NHESS paper, which should be considered as selfplagiarism if not cited properly.

Reply: We have considered that many readers do not have time to read Part 1, thus we added basic information of the catchment and dam by citing Part 1 paper. But now the paper is restructured considering your comments and keeps balance with Part 1.

**Changes in manuscript:**

We summarize our changes in the revised version:
- Section 1 Introduction now emphasizes this study is a continuation of the Part 1 in a series on the hydrological forecast for the Kasahori dam. The main difference is that in the Part 2 we could access a better ensemble forecast for rainfall over the domain around the Kasahori dam. An interesting feature of this ensemble forecast is a large number of ensemble members which suggested us many interesting problems in verification. Some new references have also been updated in this section.
- Section 2 on the heavy rainfall event has been rewritten. We have mainly referred to the Part 1 for necessary information. A text has been added to briefly explain the mechanism that causes heavy rain in the area around the Kasahori dam.
- Section 3 on meteorological ensemble forecast has been divided into two subsections: the subsection 3.1 uses the original content of Section 3 in the old manuscript; and the subsection 3.2 devotes to a new content on rainfall verification for the new ensemble forecast.
- Section 4 is in fact the original Section 5 in the old manuscript. We have removed the original Section 4 on the Kasahori dam catchment since all information is available from the Part 1.
- Section 5 on the results has been divided into two subsections: the subsection 5.1 presents probabilistic forecasts for hydrological variables; and the subsection 5.2 describe a theory on selection of best members based on past performance and its application in our case. Whereas the theory is the new content, the application is mainly based on the content in the old manuscript. With this theory, a call for machine learning methods has been irrelevant now and all texts involving this topic has been removed.
- Section 6 Conclusion has been slightly modified by adding discussion on the possibility of

introducing spatial and temporal uncertainty into NSE.

[revised manuscript text omitted]

---

## Referee Report (RR1)

**Referee report**

nhess-2018-343 Submitted on 13 Nov 2018 Companion Paper nhess-2015-271

Ensemble flood simulation for a small dam catchment in Japan using nonhydrostatic model rainfalls. Part 2: Flood forecasting using 1600 member 4D-EnVAR predicted rainfalls

Kenichiro Kobayashi, Le Duc, Apip, Tsutao Oizumi, and Kazuo Saito

**General comments:**

The authors provide an interesting study about using ensemble forecasts with a very large number of members and subsequent reduction of the members for a more "practical" application in real time flood forecasting. The temporal and spatial error of NWP is less investigated than the error in the predictor. This paper addresses such issues and is of scientific interest.

The study is based on only one flood event, which makes conclusions very difficult. It is a major drawback, but an inherent problem in using a relatively new technique for forecasting very rare events. Therefore, I consider studies about single events useful for the flood forecasting community even if there should not be a significant advance of science. However, this must be made clearer.

Other authors have already done a lot of research in ensemble calibration or member reduction techniques for application in forecasting. E.g., methods based on Bayesian theory and regression techniques were used to assign weights to the members of an ensemble forecast based on prior information. Here, the authors should extend their literature study in their introduction (Reich and Cotter is "just" a text book citation). They could also add this aspect to the objectives of their work.

How stable is the selection of members with time? It seems to be relatively stable for their case study, but might happen that the selection must be updated very often. I have doubts if a single event study can provide a solution ready for operational forecasting. In the discussion, they should give more information or judgement if the proposed solution is transferrable to other events and make limitations clearer.

**Specific comments:**

Abstract: when introducing the "dynamical selection" of the best ensemble members (a kind of subensemble), the authors should not refer to the criterion (NSE), which can be questioned, but mention the techniques applied. Furthermore, they should make clear that only a single extreme event was used for the study.

P 1 L 25ff: ensemble forecasts do not necessarily give the probability of occurrence of a flood – that is a common misunderstanding. A good ensemble gives information about the range of uncertainty ("frames the future development"), i.e. the observation should be within the uncertainty band. Most ensemble forecasts assume the same probability for each member and use frequency evaluations. Probability is obtained by data assimilation and prost-processing, as the authors have added in their revision. The authors could carefully revise their usage of probability in the text and check where frequency is a more appropriate term.

P 3 L 27: "The main theme of this Part 2 paper is that the 1600 ensemble rainfall forecasts can significantly improve the rainfall forecast over the large area around Kasahori dam": They should not give a theme with statements of their result, but first put the research question and objectives here. Also, after the revision, they have added work regarding the selection of best members in an operational case. It could be mentioned now in the objectives if not the title.

Section 2 is a very short – the content might be moved to section 1.

P 5 L 31: The rationale of the FSS should be briefly explained. Please explain the meaning of high and low values (just add sth. like "can have values between x and y, where y indicates the best possible score..."). Equations can be referred to by the citation. The reference (Duc et al., 2013) is not appropriate. The FSS is relatively new, so the original source must be cited here instead of own work of the authors using the FSS, which others proposed earlier.

P 5 L 34 "Note that an additional experiment with 4DEnVAR-NHM using 50 ensemble members": how were the 50 members produced? Please give information (or a citation) about the differences in the ensemble generation mechanism of the 50- and 1600-member ensembles.

P 6 L 26: "but the ensemble mean precipitation is smeared out as a side effect of the averaging procedure": then, the averaging procedure is maybe not a good solution. This is a well-known effect of ensembles of large size. Instead of averaging, other authors have used ensemble size reduction techniques. Finally, the authors did that but do not introduce at this stage of the manuscript (see comment above).

P 7 I 26 ff: The calibration of a hydrological model for a single event is questionable. Furthermore, using radar data instead of observations rises questions about the quality of the radar data, as can be seen later (fig. 13). Calibrating against "wrong" inputs produces higher uncertainty of the hydrological model, because it does not represent the physical processes well. The observed runoff is not a product of the radar data but a product of the observed rain. Parameters could get non-behavioral values in order to fit with the wrong rain input. As the authors assume a perfect hydrological model (without considering its uncertainty), it should be calibrated against the most perfect input data available. I think that the hydrological model is not valid here. However, the calibration and their discussion could be updated with a reasonable effort and the overall study is not about hydrology. If the radar data are used in operational service, but an error is known, the input data must be improved or post-processing can be applied, e.g. bias corrections. Research went a lot further in these topics.

P 8 L 7: In figure 4, it can be seen that the observation captured all three peaks quite well. Observed data show a consistent behavior. The hydrological model shows weakness in simulating double-peak flood waves. This would not necessarily prohibit it's use for the study. The authors could add observed areal rainfall in Fig. 5 for better interpretation. From fig. 13, it is clear that there is an underestimation of rainfall by the radar products compared to the gauge stations. It would be good to use observed rain input to simulate stream flow as the reference for comparisons.

P 8 L 29: "observed rainfall within the range" – I think that Fig. 6 only shows runoff ensembles, so instead of "rainfall" they should use "runoff" here.

P 9 L 2 ff: see comment for fig. 7.

P 10 L 17 ff.: The authors should add if the NSE is of stream flow. Why not choose the best 50 members of the rain forecast?

Fig. 1: what is the meaning of the spatial scale? I did not find an explanation in the text or the caption.

Fig. 2: Y axis is "Observed Relative Frequency" and should be labelled accordingly. The reliability diagram is not intuitive, in my opinion not even appropriate for a single event situation – even if recommended by one reviewer. Usually, there are not such pairs like 90 % forecast probability and 0% observed frequency. For small data sets, other authors have used bootstrapping to refine the probability distributions. However, from a single event, one cannot conclude the reliability of a certain forecast technique. The authors may re-consider their usage of reliability diagrams in that

context. Fig. 1 gives a good idea of the performance of the different systems for different rain intensities. I propose to leave out the reliability diagrams here, or replace with a more suitable performance measure for the single event. Maybe just give the Brier score or other metrics, as mentioned by reviewer #2.

Fig. 4: does R/A stand for "observation"? Is that radar composite or ground based? The plot is not easy to understand. It would be better to draw the 5-95 percentiles as light gray, the iqr as darker gray areas and the observed/derived lines as such, in colors. As done in fig. 8.

Fig. 6: Using the same style as figure 8 would be more informative. Then, figure 8 could be left out.

Fig. 7: the NSE is not good in characterizing the performance of stream flow ensembles of a single event. It is usually applied for calibrating hydrological models against observations, and more common for long time series. I propose to leave out fig. 7, and remove the corresponding text. It does not add to the findings but rises questions.

Fig. 10: again, the style of fig. 8 would be better here.

---

## Referee Report (RR2)

In the previous two reviews, I strongly suggested validation at multiple locations because capability of robust flood forecasts cannot be claimed from the result at a single streamflow gauge location. However, rigorous verification at multiple locations is not included in the latest revision. Except for the use of large ensemble, unfortunately, it is hard to find novelty in this manuscript. Especially, the proposed method for dynamic selection of ensemble cannot be considered as a new approach because it just follows the basic procedures of particle filtering which was introduced in hydrology 15 years ago (Moradkhani et al., 2005; Weerts and El Serafy, 2006). The capability of this selection method is not also demonstrated properly. In the authors' response, they argued that "the new method is established under the mathematical framework of particle filtering and data assimilation in general and the theory here does not depend on its application for any specific location". This is the argument which should be demonstrated with clear evidence. Compared to their previous companion paper in NHESS, however, the study domain for flood forecasts rather reduced from two catchments to one. When I briefly searched for availability of streamflow observations from Japanese websites (e.g. www.river.go.jp see the figure below), it turns out that there are a lot of streamflow gauges and dam reservoirs in the impacted area. Therefore, I am not positive that any readers can have meaningful implications from the limited analysis shown in the manuscript. Without additional evidence, I do not recommend the final publication. I think the form of discussion paper without final publication would be appropriate for this result.

[Figure]

Fig 3. Mean precipitation

Streamflow observation gauges

References

Moradkhani, H., Hsu, K.-L., Gupta, H., Sorooshian, S., 2005. Uncertainty assessment of hydrologic model states and parameters: Sequential data assimilation using the particle filter. Water Resour. Res. 41. https://doi.org/10.1029/2004WR003604

Weerts, A.H., El Serafy, G.Y.H., 2006. Particle filtering and ensemble Kalman filtering for state updating with hydrological conceptual rainfall-runoff models. Water Resour. Res. 42. https://doi.org/10.1029/2005WR004093

---

## Author Response (AR2)

Dear Editor and Reviewers,

Thank you very much again for the time you spent for the review of our paper. Considering your comments carefully, we would like to answer as follows.
* * *
**Reviewer 3**
General comments:

The authors provide an interesting study about using ensemble forecasts with a very large number of members and subsequent reduction of the members for a more "practical" application in real time flood forecasting. The temporal and spatial error of NWP is less investigated than the error in the predictor. This paper addresses such issues and is of scientific interest.The study is based on only one flood event, which makes conclusions very difficult. It is a major drawback, but an inherent problem in using a relatively new technique for forecasting very rare events. Therefore, I consider studies about single events useful for the flood forecasting community even if there should not be a significant advance of science. However, this must be made clearer.

Reply: We have revised the Introduction to make the object of this study more clear, especially to emphasize that this study is a continuation of Part 1 for the Niigata-Fukushima heavy rainfall event with an updating in the rainfall ensemble forecasts to use. New methods are required to deal with the large number of ensemble members in this case, which makes Part 2 more interesting. We have added the following paragraphs into the Introduction:

Recently, as a further improvement upon the 2 km downscale ensemble rainfall simulations used by Kobayashi et al. (2016), Duc and Saito (2017) developed an advanced data assimilation system with the ensemble variational method (EnVAR) and increased the number of ensemble members to 1600. This new data assimilation system was aimed to improve the rainfall forecasts of the 2011 Niigata-Fukushima heavy rainfall event. The torrential rain of this event occurred over the small area along the synoptic scale stationary front (for surface weather map, see Fig. 1 of Kobayashi et al. 2016). Saito et al (2013) found that the location where intense rain concentrates varied to small changes of the model setting, thus the position of the heavy rain was likely controlled by horizontal convergence along the front, rather than the orographic forcing.

Since the new EPS produced better forecasts of the rainfall fields, in this study, as a Part 2 version of Kobayashi et al. (2016), we applied those 1600 ensemble rainfalls to the ensemble inflow simulations to Kasahori Dam. In the series of Part 1 and 2, we intentionally have chosen a rainfall-runoff model whose specification is quite close to those runoff models used in many governmental practices of Japanese flood forecasting to see the usefulness of 1600 ensemble rainfalls. Our objective is to assess impact of the improvement of the rainfall forecast over the large area around Kasahori dam on the streamflow forecast for the Kasahori dam. In Part 1 the technique of positional lag correction has been applied to match the rainfall forecasts with the observations to have a better hydrological forecast. This technique is hard to be applied in real-time flood forecasting since rainfall observations are unknown and there exist a lot of potential positional lag vectors to choose. Statistically the positional lag vector should respond to the local orographic features but it may vary depending on the synoptic condition on the day and model forecast errors in a specific event. Thus the positional lag vector for one extreme rainfall event basically cannot be applied to other extreme event as is. The new EPS is expected to remove the use of such technique.

In addition, the very large number of ensemble members, which is 10 to 20 times larger than the typical number of ensemble members currently run in operational forecast centers, poses new issues needed to solve in computation and interpretation. First, regional forecast centers may not afford running 1600 hydrological forecasts in real-time and a method to choose the most important members may be helpful. Such kind of method is known as ensemble reduction in ensemble forecast (Molteni et al., 2011; Montani et al., 2011; Hacker et al., 2011; Weidle et al., 2013; Serafin et al., 2019), which is built upon cluster analysis when observations are not used as a guidance for selection. However, our problem is more interesting where we can access the observations at the first few hours and ensemble reduction should make use of these past observations in selecting important members. Second, it is more challenging to interpret the result when temporal and

spatial uncertainties are realized more distinct now. Without taking such uncertainties into account, the ensemble forecasts are easily to be considered as useless.

Other authors have already done a lot of research in ensemble calibration or member reduction techniques for application in forecasting. E.g., methods based on Bayesian theory and regression techniques were used to assign weights to the members of an ensemble forecast based on prior information. Here, the authors should extend their literature study in their introduction (Reich and Cotter is "just" a text book citation). They could also add this aspect to the objectives of their work.

Reply: We have added citations to the studies on ensemble reduction in the Introduction as shown in the previous reply. Also citations to the data assimilation literature have been extended in Section 4.2 to emphasize our approach in deed roots into the data assimilation theory.

How stable is the selection of members with time? It seems to be relatively stable for their case study, but might happen that the selection must be updated very often. I have doubts if a single event study can provide a solution ready for operational forecasting. In the discussion, they should give more information or judgement if the proposed solution is transferrable to other events and make limitations clearer.

Reply: Thank you for this interesting question. In fact Figure 14 (Figure 12 in revised manuscript) shows that the set of the best members are not stable at all. We have added a paragraph to explain this behavior. The limitation of our approach is explained in the next paragraph. We reproduce the two paragraphs here:

It is very clear from Figure 12 that the set of the best members varies considerably with the time intervals of available observations. This is because the NSE index is sensitive to the large difference between forecasts and observations. That means, unless we simulate all the discharges of the 1600 members in advance, we may need to run many new members to update this set every time when new observations are available and this causes management of the best members more complicated. To see why this occurs, suppose that we have a member with the sums $\sum_{i=1}^{N-1}\{Q_0^i - Q_s^i\}^2$ are almost zero for the first 1, 2, ..., N-1 hours when we only have no rain or light rain during this time. When we consider the next hour to reselect the best members, if the term $\{Q_0^N - Q_s^N\}^2$ becomes very large, this member will suddenly be out of favour despite the fact that it is always selected as one of the best members in all previous selection rounds. However, this large difference may come from spatial and temporal displacement errors of rainfall forecasts and not necessary reflect an inaccurate forecast. This shows that the use of NSE in selecting the best members is quite sensitive to spatial and temporal displacement errors of rainfall. Part 1 of this study is an illustration for impact of spatial displacement errors on forecast performance while Figures 7 and 9 here show the case of temporal displacement errors. On the other hand, NSE of rainfall cannot be used to select the best discharge members since rainfall NSEs of similar values can produce different discharge hydrographs. For example, the catchment average rainfall with NSE of around 0 produces discharges with NSE close to 0.5 and -0.5. The spatial distribution of the rainfall field causes these differences even though the amount of the catchment average rainfalls is the same. Even if the catchment area is small, different patterns in the rainfall field bring different discharge simulations with different NSEs. Furthermore, the error model for rainfall does not follow the Gaussian distribution and a more appropriate distribution like gamma or lognormal should be used. However, such distributions make NSE irrelevant and new verification scores derived from these distributions are needed, which can take the form like FSS. Thus, it is expected that if we can introduce spatial and temporal uncertainty in modelling the likelihood $p_Y(\mathbf{y}|\mathbf{x}_i)$, the predictive pdf (6) could yield a more useful ensemble forecast. However, this requires a lengthy mathematical treatment that is worth to explore in details in a separate study.

Specific comments:
Abstract: when introducing the „dynamical selection" of the best ensemble members (a kind of sub-ensemble), the authors should not refer to the criterion (NSE), which can be questioned, but mention the techniques applied. Furthermore, they should make clear that only a single extreme event was used for the study.

Reply: We have updated the abstract following your suggestion. The revised abstract is as follows:

Abstract. This paper is a continuation of the authors' previous paper (Part 1) on the feasibility of ensemble flood forecasting for a small dam catchment (Kasahori dam; approx.70 km2) in Niigata Japan using a distributed rainfall-runoff model and rainfall ensemble forecasts. The ensemble forecasts were given by an advanced data assimilation system, a four-dimensional ensemble variational assimilation system using the Japan Meteorological Agency non-hydrostatic model (4D-EnVAR). A noteworthy feature of this system was the use of a very large number of ensemble members (1600), which yielded a significant improvement in the rainfall forecast compared to Part 1. The ensemble flood forecasting using the 1600 rainfalls succeeded in indicating the necessity of emergency flood operation with the occurrence probability and enough lead time (e.g., 12 hours) with regard to this extreme event. A new method for dynamical selection of the best ensemble members  based on the Bayesian reasoning with different evaluation periods is proposed. As the result, it is recognized that the selection based on Nash Sutcliffe Efficiency does not provide an exact discharge forecast with several hours lead time, but it can provide some trend in the near future.

P 1 L 25ff: ensemble forecasts do not necessarily give the probability of occurrence of a flood – that is a common misunderstanding. A good ensemble gives information about the range of uncertainty ("frames the future development"), i.e. the observation should be within the uncertainty band. Most ensemble forecasts assume the same probability for each member and use frequency evaluations. Probability is obtained by data assimilation and post-processing, as the authors have added in their revision. The authors could carefully revise their usage of probability in the text and check where frequency is a more appropriate term.

Reply: We have updated the first sentence of the Introduction to show benefit of ensemble forecast:
Flood simulation driven by ensemble rainfalls has attracted more attention in recent years with a lot of useful information that ensemble flood forecasts can provide in flood control such as forecast uncertainty, probabilities of rare events, and potential flooding scenarios.

Each ensemble forecast can be identified with an empirical pdf

$$p_X(\mathbf{x}) = \sum_{i=1}^{K} \frac{1}{K} \delta(\mathbf{x} - \mathbf{x}_i), \qquad (1)$$

as we showed in Equation (3) in the text. It is clearly that the above pdf assume "the same probability for each member" as the reviewer said. So from this pdf if we want to know the probability that x is greater a certain threshold we can integrate this pdf and due to its Dirac mixture nature, the probability reduces to counting the number of ensemble members that have their forecasts greater than this threshold, which in other words the frequency. Therefore, we think that both the uses of probability or frequency are valid here.

P 3 L 27: "The main theme of this Part 2 paper is that the 1600 ensemble rainfall forecasts can significantly improve the rainfall forecast over the large area around Kasahori dam": They should not give a theme with statements of their result, but first put the research question and objectives here. Also, after the revision, they have added work regarding the selection of best members in an operational case. It could be mentioned now in the objectives if not the title.

Reply: We have removed this sentence and updated the Introduction in accordance with your suggestion as mentioned in the reply to the general comments.

Section 2 is a very short – the content might be moved to section 1.

Reply: Modified.

P 5 L 31: The rationale of the FSS should be briefly explained. Please explain the meaning of high and low values (just add sth. like "can have values between x and y, where y indicates the best possible score…"). Equations can be referred to by the citation. The reference (Duc et al., 2013) is not appropriate. The FSS is relatively new, so the original source must be cited

here instead of own work of the authors using the FSS, which others proposed earlier.

Reply: The reference to the original paper has been done. We have added the following paragraph to briefly explain the rationale underlying the FSS:

Figure 1 shows the verification results for the 3-hour precipitation forecasts as measured by the Fraction Skill Score (FSS) (Roberts and Lean, 2008). Given a rainfall threshold and an area around a grid point, which is called a neighborhood, the FSS indicates relative difference between observed and forecasted rainfall fractions in this area. This verification score is used to mitigate difficult in rainfall verification at grid scale with very high-resolution forecasts in which high variability of rain fields usually makes the traditional scores inadequate due to their requirement of exact match between observations and forecasts at grid points. Thus the solution that the FSS follows is to consider forecast quality at spatial scales coarser than grid scale by comparing forecasts and observations not at grid points but at neighbourhoods whose sizes are identified with spatial scales. The FSS is normalized to range from 0 to 1 with the value of 1 indicating a perfect forecast and the value of 0 a no-skill forecast which can be obtained by a random forecast.

P 5 L 34 "Note that an additional experiment with 4DEnVAR-NHM using 50 ensemble members": how were the 50 members produced? Please give information (or a citation) about the differences in the ensemble generation mechanism of the 50- and 1600-member ensembles.

Reply: We have added the sentence below to provide information for this additional experiment:

The main difference between 4DEnVAR50 and 4DEnVAR1600 is that vertical localization was applied in the former case to generate its ensemble members. As mentioned in the previous section, vertical localization can potentially weaken vertical flows in convective areas by removing physically vertical correlations.

P 6 L 26: "but the ensemble mean precipitation is smeared out as a side effect of the averaging procedure": then, the averaging procedure is maybe not a good solution. This is a well-known effect of ensembles of large size. Instead of averaging, other authors have used ensemble size reduction techniques. Finally, the authors did that but do not introduce at this stage of the manuscript (see comment above).

Reply: We have added some sentences in Section 2 to highlight this problem and show a potential solution developed later in Section 4:

Therefore, the ensemble mean should not be used into our hydrological model as a representative of the ensemble forecast. Rather than that, all ensemble precipitation forecasts should be fed into the hydrological model to avoid rainfall distortion caused by averaging in addition to a faithful description of rainfall uncertainty. Of course with 1600 members this causes a huge increase in computational cost and we will try to reduce this burden by testing a suitable dynamical selection described later in Section 4.

P 7 l 26 ff: The calibration of a hydrological model for a single event is questionable. Furthermore, using radar data instead of observations rises questions about the quality of the radar data, as can be seen later (fig. 13). Calibrating against "wrong" inputs produces higher uncertainty of the hydrological model, because it does not represent the physical processes well. The observed runoff is not a product of the radar data but a product of the observed rain. Parameters could get non-behavioral values in order to fit with the wrong rain input. As the authors assume a perfect hydrological model (without considering its uncertainty), it should be calibrated against the most perfect input data available. I think that the hydrological model is not valid here. However, the calibration and their discussion could be updated with a reasonable effort and the overall study is not about hydrology. If the radar data are used in operational service, but an error is known, the input data must be improved or post-processing can be applied, e.g. bias corrections. Research went a lot further in these topics.

Reply: We wrote in Part 1 paper that the catchment-averaged cumulative rainfalls during the period were 765.0 mm for

ground rain-gauges (RG), 762.8 mm for Radar AMEDAS (RA, operational precipitation analysis of JMA based on radar and rain-gauge observations), and 568.5 mm for Radar-Composite (RC), respectively. Thus, the cumulative rainfall by RC is 0.74 times that of RG, whereas the value by RA is almost similar to RG. Considering your comment, we decided to replace all the figures using parameters by RC (in previous version paper) with those figures using parameters by RA. However, in the paper we kept several figures by the parameters with RC as references since it can show the forecast uncertainty by the model parameters. Figures 5-12 (in revised manuscript) are all replaced by those with RA parameters. Likewise, the sentence below was added.

The parameters of the DRR model were recalibrated in this study using hourly Radar AMEDAS since the amount of total rainfall for the period (762.8mm) is closer to ground rain-gauge (765.0 mm) (Kobayashi et. al., 2016). The hourly Radar-Composite (RC, radar data) of JMA was also used for another recalibration as a reference since Radar precipitation data is in general the primary source for real time flood forecasting. The total rainfall amount with RC was 568.5 mm which is smaller than the ground rain-gauge (765.0 mm).

P 8 L 7: In figure 4, it can be seen that the observation captured all three peaks quite well. Observed data show a consistent behavior. The hydrological model shows weakness in simulating double-peak flood waves. This would not necessarily prohibit it's use for the study.

Reply: The simulated hydrograph with RA shows better reproduction especially for the 2nd peak in Figure 5 (in revised manuscript), though NSE of the 72 hour hydrograph with RA (NSE=0.686) is not necessarily better than that with RC (0.743). Considering your comment, in the paper we decided to show the results basically with the parameters by RA since the rainfall is considered more accurate and, as reviewer mentioned, the model parameters are physically more meaningful. After all, the sentence below was added.

In the calibration simulations in Figure 5, the NSEs with RA and RC are 0.686 and 0.743 respectively. Although the NSE with RA is worse than RC, the total rainfall amount with RA is considered more accurate and the 2nd and 3rd discharge peaks seem to be captured better with RA, thus the following discussion will be made basically with the parameters calibrated with RA. Some results with RC will be added as references. The main difference of the parameters between RA and RC is that the surface soil thickness D to hold the rainfall at the initial stage is thicker in RA, which yields the lower discharge in the river.

The authors could add observed areal rainfall in Fig. 5 for better interpretation. From fig. 13, it is clear that there is an underestimation of rainfall by the radar products compared to the gauge stations. It would be good to use observed rain input to simulate stream flow as the reference for comparisons.

Reply: We decided to show the results with the model parameters by RA, then we show some results with the parameters by RC (previous version). Likewise the areal rainfall is added in Fig. 5.

P 8 L 29: "observed rainfall within the range" – I think that Fig. 6 only shows runoff ensembles, so instead of "rainfall" they should use "runoff" here.

Reply: Thank you very much. Modified.

P 9 L 2 ff: see comment for fig. 7.

Reply: Figure 7 is removed considering your comment.

P 10 L 17 ff.: The authors should add if the NSE is of stream flow. Why not choose the best 50 members of the rain forecast?

Reply: We have added a paragraph to explain why it is more problematic if we want to choose the best 50 rainfall forecasts

using NSE. The paragraph is as follows.

This shows that the use of NSE in selecting the best members is quite sensitive to spatial and temporal displacement errors of rainfall. Part 1 of this study is an illustration for impact of spatial displacement errors on forecast performance while Figures 7 and 9 here show the case of temporal displacement errors. On the other hand, NSE of rainfall cannot be used to select the best discharge members since rainfall NSEs of similar values can produce different discharge hydrographs. For example, the catchment average rainfall with NSE of around 0 produces discharges with NSE close to 0.5 and -0.5. The spatial distribution of the rainfall field causes these differences even though the amount of the catchment average rainfalls is the same. Even if the catchment area is small, different patterns in the rainfall field bring different discharge simulations with different NSEs. Furthermore, the error model for rainfall does not follow the Gaussian distribution and a more appropriate distribution like gamma or lognormal should be used. However, such distributions make NSE irrelevant and new verification scores derived from these distributions are needed, which can take the form like FSS. Thus, it is expected that if we can introduce spatial and temporal uncertainty in modelling the likelihood $p_Y(\mathbf{y}|\mathbf{x}_i)$, the predictive pdf (6) could yield a more useful ensemble forecast. However, this requires a lengthy mathematical treatment that is worth to explore in details in a separate study.

Fig. 1: what is the meaning of the spatial scale? I did not find an explanation in the text or the caption.

Reply: We have explained the meaning of spatial scales in a short description for the rationale underlying the FSS. The description is as follows.

Figure 1 shows the verification results for the 3-hour precipitation forecasts as measured by the Fraction Skill Score (FSS) (Roberts and Lean, 2008). Given a rainfall threshold and an area around a grid point, which is called a neighborhood, the FSS indicates relative difference between observed and forecasted rainfall fractions in this area. This verification score is used to mitigate difficult in rainfall verification at grid scale with very high-resolution forecasts in which high variability of rain fields usually makes the traditional scores inadequate due to their requirement of exact match between observations and forecasts at grid points. Thus the solution that the FSS follows is to consider forecast quality at spatial scales coarser than grid scale by comparing forecasts and observations not at grid points but at neighbourhoods whose sizes are identified with spatial scales. The FSS is normalized to range from 0 to 1 with the value of 1 indicating a perfect forecast and the value of 0 a no-skill forecast which can be obtained by a random forecast.

Fig. 2: Y axis is "Observed Relative Frequency" and should be labelled accordingly. The reliability diagram is not intuitive, in my opinion not even appropriate for a single event situation – even if recommended by one reviewer. Usually, there are not such pairs like 90 % forecast probability and 0% observed frequency. For small data sets, other authors have used bootstrapping to refine the probability distributions. However, from a single event, one cannot conclude the reliability of a certain forecast technique. The authors may re-consider their usage of reliability diagrams in that context. Fig. 1 gives a good idea of the performance of the different systems for different rain intensities. I propose to leave out the reliability diagrams here, or replace with a more suitable performance measure for the single event. Maybe just give the Brier score or other metrics, as mentioned by reviewer #2.

Reply: We agree with the reviewer that reliability diagrams are prone to sampling errors if we populate it with a small sample of observation – forecast pairs. Although we plotted our reliability diagrams for one event, the dataset was in fact not small at all since we used 3-hour observation – forecast pairs from several intervals with the verification domain covering not only the Kasahori dam but also a more larger region around Fukushima and Niigata prefectures. Also the Brier scores are in fact given implicitly in the reliability diagrams since each Brier score can be decomposed as
Brier score = Reliability – Resolution + Uncertainty,    (2)
where the three terms on the right-hand side are represented in the reliability diagrams. To make them visible, in the revised version we have shown these values explicitly in the plots. Another reason to keep the reliability diagrams is that Figure 1 with the FSS only shows the performance of the deterministic forecasts but not the ensemble forecasts and the reliability diagrams are needed if we want to access the performance of the ensemble forecasts.

Fig. 4: does R/A stand for "observation"? Is that radar composite or ground based? The plot is not easy to understand. It would be better to draw the 5-95 percentiles as light gray, the iqr as darker gray areas and the observed/derived lines as such, in colors. As done in fig. 8.

Reply: Modified.

Fig. 6: Using the same style as figure 8 would be more informative. Then, figure 8 could be left out.

Reply: Modified.

Fig. 7: the NSE is not good in characterizing the performance of stream flow ensembles of a single event. It is usually applied for calibrating hydrological models against observations, and more common for long time series. I propose to leave out fig. 7, and remove the corresponding text. It does not add to the findings but rises questions.

Reply: Removed accordingly.

Fig. 10: again, the style of fig. 8 would be better here.

Reply: Modified.

**Reviewer 2**

I express my gratitude to the authors for their detailed response and the revised manuscript. I carefully read them to provide my suggestion. The main reasons for suggesting reject & resubmit in the previous review include 1) that apparently probabilistic discharge forecasts by a large set of ensemble cannot be seen as an improved prediction and 2) the proposed ensemble selection method has a significant potential flaw because the method is valid only for one location and not feasible for distributed modeling. To resolve these issues, I suggested validation of discharge simulation at multiple gaging locations within the study catchment and rigorous evaluation with probabilistic measures. Despite the efforts of the authors, I don't think these two main issues are properly addressed in the revised manuscript. Therefore, I do not recommend the publication of this manuscript in NHESS.

If this article is published despite my opposition, I suggest the readers and editors take a close look at 1600 ensemble discharge traces shown in gray (Fig 6) which fill a whole range of discharge values from 0 to 2,000 cms. I don't think any expert can make a meaningful decision on the risk management with these highly uncertain forecasts even though the mean values are close to the observed. Unfortunately, in my view, this is one of the typical cases which should be avoided in the probabilistic forecasts. However, in this study, rigorous statistical evaluations (e.g. calculation of probabilistic scores per each forecast lead time throughout the simulation) are lacking and failed to demonstrate the adequacy of the probabilistic discharge forecasts.

As a remedy for too dispersed ensemble forecasts, an ensemble selection method adapted from particle filtering was applied in the latter part of the revised manuscript. However, in addition to the fact that the performance of the selected ensemble deteriorates quickly, there is a critical flaw that this selection may be valid only for one location. To resolve this issue, the evaluation is required at multiple locations, which is also lacking in the manuscript.

Reply: The first argument of the reviewer for rejection of our research is the claim that the probabilistic discharge forecasts as shown in Figure 6 cannot be seen as a good forecast. Here the reviewer assumed that the pdf at each forecast lead time is a uniform distribution ranging from 0 to 2000 m3/s. However, this is only an illusion caused by plotting since any curve crosses this forecast time will leave a spot with gray color. The true distributions should be read from Figure 8 (Figure 6 in revised manuscript) which shows the interquartile ranges are indeed much smaller than 2000 m3/s. As we discussed many

times in the text, Figure 8 (Figure 6 in revised manuscript) still leads to misjudgment for the true performance of the ensemble forecast the temporal displacement errors were not considered here, which is inherently the same problem in rainfall verification that leads to the development of new verification scores like FSS. To mitigate impact of temporal displacement errors, it is better to consider the accumulated volume forecasts as shown in Figures 10 and 11 (Figures 8 and 9 in revised manuscript). Figure 11 (Figure 9 in revised manuscript) itself shows the value of the ensemble forecast in a visual way rather than a score like the Brier score, which is indeed, the sum of differences between two curves in Figure 11.

The second argument for rejection is the claim that our new ensemble reduction method has a flaw that it is only valid for one location. We disagree with this claim due to the fact that the new method is established under the mathematical framework of particle filtering and data assimilation in general. Thus, the theory here does not depend on its application for any specific location. We should differentiate here the theoretical aspect of a method with its application. To use this method in practice, the choice of the error model for observations plays a crucial role here. In the paper, we chose the Gaussian model which is equivalent to the use of NSE to select the best members. This choice somehow works in the paper but as we discussed it is better if we can incorporate temporal and spatial uncertainties in the error model for discharges.

[revised manuscript text omitted]

**Hourly Rainfall Forecast**
Exp: Japan02km, Ensemble members: 1600

Legend:
- ▲ R/A
- ● Ensmean
- ■ Control
- ★ Best member

Y-axis: Average Rainfall (mm/hr)

X-axis: Japanese Time (hour)

Initial date: 2011/07/29 – 00:00 JST          Forecast range: 27h

[Figure]

**Figure 4. Time series of one-hour accumulated rainfall over the catchment as forecasted by all ensemble members. The observation, control forecast, ensemble mean forecast, and best member forecast are also plotted for comparison. Here, the best member is defined as the member that has the minimum distance between its time series and the observed time series.**

[Figure]

**Figure 5.** RA hyetograph, observed dam inflow and simulated inflows with RA and RC.

**(a) 11 member**

[Figure]

**(b) 11 member (position shift)**

[Figure]

**(c) 50 member**

[Figure]

**(d) 1600 member**

[Figure]

[Figure]

**Figure 6. Hydrographs of (a) 11 discharge simulations in Part 1 (Kobayashi et al., 2016), (b) same 11 member but with a positional shift in Part 1, (c) 50 discharge simulations with 4D-EnVAR-NHM , (d) 1600 discharge simulations with 4D-EnVAR-NHM and model parameters by RA, (e) 1600 discharge simulations with 4D-EnVAR-NHM and model parameters by RC.**

[Figure]

**Figure 7. Histograms of NSE based on the simulated and observed discharges to Kasahori dam: (a) 11 discharge simulations in Part 1 (Kobayashi et al., 2016), (b) same 11 member but with a positional shift in Part 1 (Kobayashi et. al. 2016), (c) 50 discharge simulations with 4D-EnVAR-NHM and (d) 1600 discharge simulations with 4D-EnVAR-NHM.**

[Figure]

**Figure 8. The 95% confidence limits and inter-quartile limits of the 1600 ensemble members.**

[Figure]

[Figure]

**Figure 97. Probability that the simulated inflow is beyond 140 m³/s considering temporal uncertainty.**

[Figure]

**Figure .** Cumulative dam inflow by the ensemble simulations, mean of simulation and observations, as well as critical dam volume. (a) with parameters by RA, (b) with parameters by RC.

[Figure]

Accumulated Volume Forecast Probability

Critical volume: 8700000 m³

[Figure]

**Figure 9.** Probability that the dam needs emergency operation. Radar-AMEDAS and Radar composite indicate the ensemble simulation with the parameters calibrated with the rainfalls.

[Figure]

[Figure]

**Figure 1210. Hydrographs of all 1600 ensemble members, the 50 best ensemble members (NSE>0.33), and observations.**

[Figure]

**Figure 11. Rainfall intensity of the 50 best ensemble inflow simulation members, of Radar AMeDAS, of Radar-Composite, and ground observations.**

[Figure]

[Figure]

**Figure 12. (a) best 50 ensemble members (NSE >0.24) selected from first 9-hour forecast, (b) best 50 ensemble members (NSE >-0.04) selected from first 10-hour forecast, and (c) best 50 ensemble members (NSE > 0.92) selected from first 11-hour forecast.**